# A cysteine selenosulfide redox switch for protein chemical synthesis

Vincent Diemer[1], Nathalie Ollivier[1], Bérénice Leclercq[1], Hervé Drobecq[1], Jérôme Vicogne [1], Vangelis Agouridas [1✉] & Oleg Melnyk [1✉]

The control of cysteine reactivity is of paramount importance for the synthesis of proteins using the native chemical ligation (NCL) reaction. We report that this goal can be achieved in a traceless manner during ligation by appending a simple *N*-selenoethyl group to cysteine. While in synthetic organic chemistry the cleavage of carbon-nitrogen bonds is notoriously difficult, we describe that *N*-selenoethyl cysteine (SetCys) loses its selenoethyl arm in water under mild conditions upon reduction of its selenosulfide bond. Detailed mechanistic investigations show that the cleavage of the selenoethyl arm proceeds through an anionic mechanism with assistance of the cysteine thiol group. The implementation of the SetCys unit in a process enabling the modular and straightforward assembly of linear or backbone cyclized polypeptides is illustrated by the synthesis of biologically active cyclic hepatocyte growth factor variants.

[1] Univ. Lille, CNRS, Inserm, CHU Lille, Institut Pasteur de Lille, U1019-UMR 9017-CIIL-Center for Infection and Immunity of Lille, 59000 Lille, France. ✉email: vangelis.agouridas@ibl.cnrs.fr; oleg.melnyk@ibl.cnrs.fr

In recent years, the study of protein function has made tremendous advances thanks to the development of chemical synthetic tools and strategies for producing peptides and proteins. The vast majority of proteins obtained this way are assembled using native chemical ligation (NCL[1]; Fig. 1a) or derived methods[2–5]. NCL involves the reaction of a peptide thioester with a peptide equipped with an N-terminal cysteine (Cys) to produce a native peptide bond to Cys. The synthesis of complex protein scaffolds requires the control at some point of the reactivity of Cys for orienting the order by which the peptide bonds connecting the various peptide segments are produced[6] (Fig. 1a). Therefore, designing strategies for modulating Cys reactivity is a contemporary concern and stimulates the creativity of protein and organic chemists worldwide[7–11].

One hallmark of the Cys residue is its involvement in the formation of disulfide or selenosulfide bonds[12] (Fig. 1b), which often play a critical role in protein folding. Nature also exploits the redox properties of Cys thiols to control the activity of some enzymes featuring a Cys residue at their catalytic site[13]. Indeed, the conversion of a catalytic Cys thiol into a disulfide is a powerful means for shutting down enzymatic activity because disulfides are poor nucleophiles compared to thiolates. Glutathione reductase is a typical example where the enzyme becomes active upon reduction of a disulfide bond. In synthetic organic chemistry, the redox properties of the thiol group also offer a simple means for controlling its reactivity[14]. Unfortunately, acyclic dichalcogenide derivatives of Cys are labile or in fast exchange under the reducing conditions used for performing NCL. Consequently, such a bioinspired control of NCL by using Cys thiol as a redox switch has not so far proved achievable. In practice, Cys reactivity is instead masked during protein assembly by introducing classical alkyl- or acyl-based protecting groups on the α-amino group, on the side-chain thiol or both. Among the various cysteine protection strategies described so far (for a recent review, see ref. [2]), thiazolidine protecting group is certainly the most popular for protein chemical synthesis[15].

To circumvent the high lability of Cys acyclic disulfides during NCL and to use Cys thiol as a redox switch for controlling protein assembly, embedding the Cys thiol in a cyclic dichalcogenide represents a feasible solution as such species are known to be significantly less oxidizing than their linear counterparts[16]. In this work, we explore the properties of SetCys, the cyclic selenosulfide obtained by introducing a selenoethyl appendage on the α-amino group of Cys (Fig. 1c). Under NCL conditions with SetCys peptides, we show that the distribution of products vary with the strength of the reducing agent. Importantly, SetCys spontaneously loses its selenoethyl arm in water at neutral pH in the presence of popular disulfide bond reductants such as dithiothreitol (DTT) or tris(2-carboxyethyl)phosphine (TCEP). This chemical behavior contrasts with the known difficulty in breaking carbon–nitrogen bonds, a process that usually requires harsch conditions[17,18], metal catalysis[19], or radical reactions[20,21]. By contrast, the detailed mechanistic investigations reported here point toward an anionic mechanism that depends on the ionization state of SetCys in its ring-opened and reduced form. In this respect, SetCys uncovers an unusual mode of reactivity for Cys and provides a useful means for accessing complex protein scaffolds as illustrated by the total one-pot synthesis of biologically active backbone-cyclized variants of the hepatocyte growth factor (HGF) kringle 1 (K1) domain.

## Results

**SetCys peptides display redox-dependent reactivities**. The NCL reaction is classically performed in the presence of an aryl thiol catalysts[22], of which 4-mercaptophenylacetic acid (MPAA) is considered as the gold standard[23]. In addition to its catalytic abilities, the latter also contributes to the maintenance of the reactants in a weakly reducing environment. MPAA can possibly be complemented by DTT and TCEP, two powerful reductants that are popular additives for NCL. Thus, MPAA and MPAA/DTT or MPAA/TCEP additive cocktails define two extremes in reductive power applied to ligation mixtures.

We first examined the behavior of the SetCys residue in the presence of MPAA alone, i.e., weakly reductive conditions, in the search for conditions where it could be silent. Exposure of a

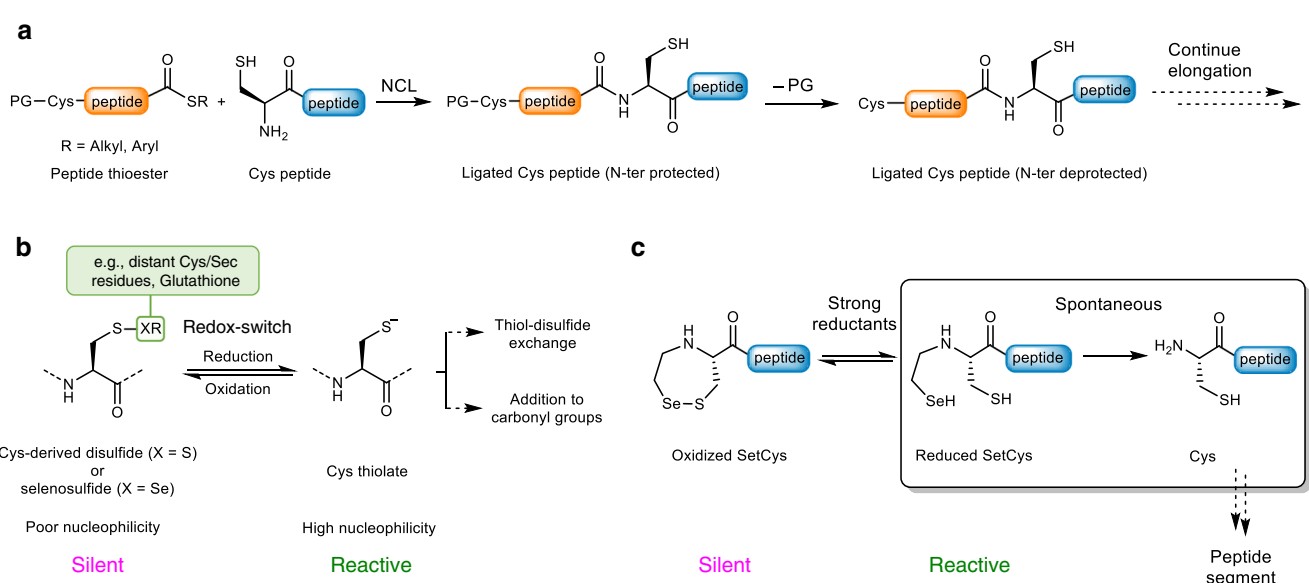

**Fig. 1 The redox control of Cys reactivity can be a mild and powerful tool for the total chemical synthesis of proteins. a** Principle of the native chemical ligation (NCL) between a C-terminal peptide thioester and a cysteinyl (Cys) peptide. The control of the site of ligation requires masking any other N-terminal Cys residue present in the mixture. **b** The reversible formation of disulfide or selenosulfide bonds is a hallmark of Cys thiol chemistry in proteins and is used by nature as a redox switch to control Cys thiol reactivity. **c** The N-(2-selenoethyl) group of SetCys shuts down the nucleophilic properties of Cys thiol by formation of a cyclic selenosulfide bond. It is removed spontaneously in water at neutral pH under strong reductive conditions.

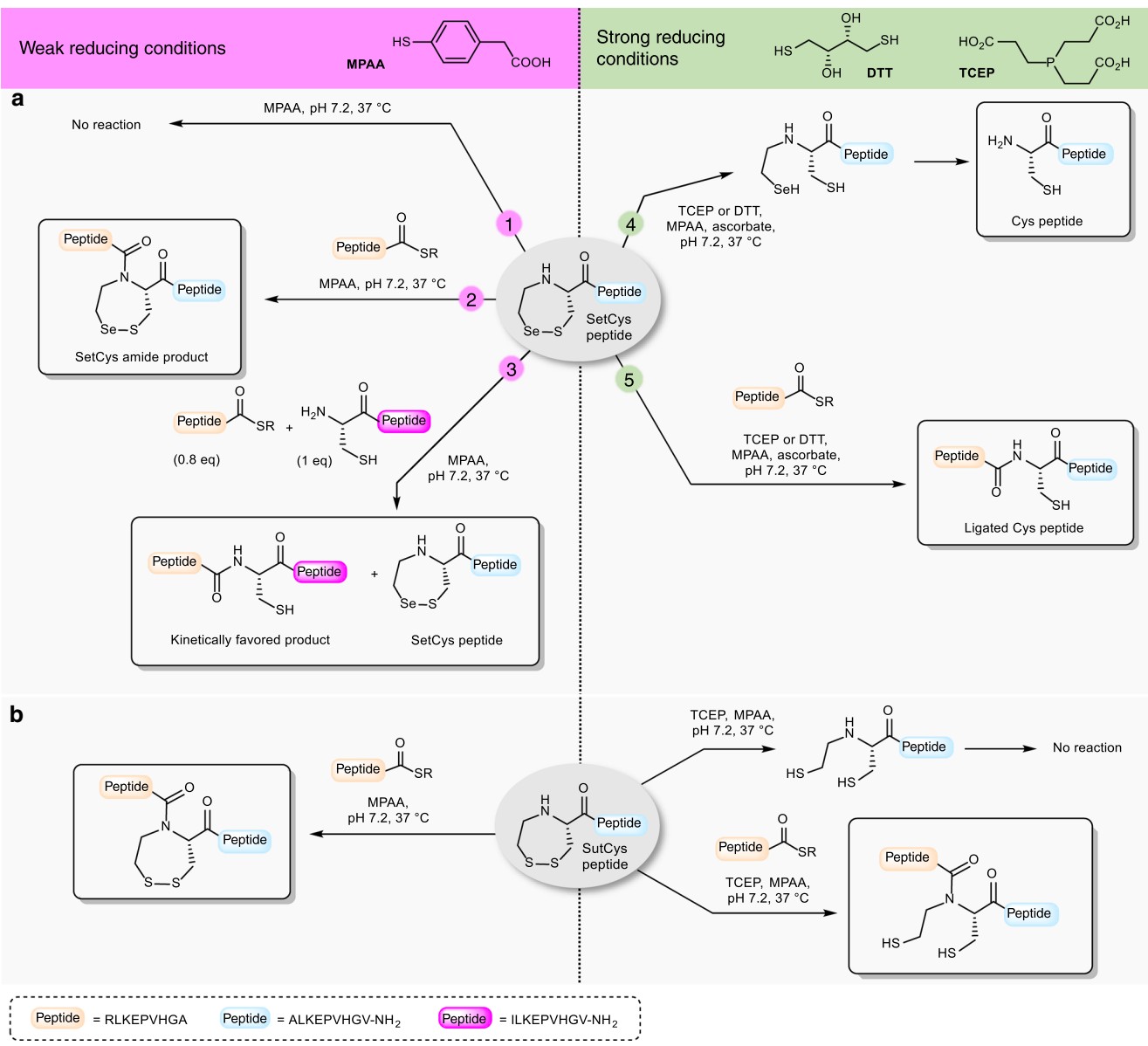

**Fig. 2 Reactivity of *N*-(2-chalcogenoethyl)cysteine (SetCys and SutCys) peptides as a function of the reducing strength of the mixture. a** The reactivity of *N*-(2-selenoethyl)cysteine (SetCys) peptides is controlled by the reducing strength of the mixture. The numbers on the arrows indicate different experimental conditions. Conditions 1–3 (−SR = 3-mercaptopropionic acid, MPA): weakly reducing conditions, typically in the presence of an excess of 4-mercaptophenylacetic acid (MPAA) at neutral pH. Conditions 4 and 5 (−SR = MPA or MPAA): strong reducing conditions, typically in the presence of dithiothreitol (DTT) or *tris*(2-carboxyethyl)phosphine (TCEP) at neutral pH. **b** Ligation of an *N*-(2-sulfanylethyl)cysteine (SutCys) peptide with a peptide alkyl thioester under weak or strong reducing conditions.

model SetCys peptide to a large excess of MPAA at neutral pH led to no apparent change (Fig. 2a, property 1; Supplementary Fig. 54). In a second experiment, incubation of the SetCys peptide with a peptide thioester in the presence of MPAA furnished a ligated peptide featuring an internal SetCys residue (Fig. 2a, property 2). Although we could not detect any reduced SetCys in the presence of MPAA alone (Fig. 2a, property 1), perhaps due to its oxidation by molecular oxygen during workup and analysis, the formation of the SetCys amide product in this experiment shows that this species is likely present under these conditions. However, the rate of ligation was more than 10-fold lower than the rate observed for NCL with a Cys peptide. This observation prompted us to run a competitive reaction in which a peptide thioester and an equimolar mixture of SetCys and Cys peptides were reacted in the presence of MPAA (Fig. 2a, property 3).

Interestingly, this experiment resulted in the exclusive formation of the ligation product with the Cys peptide. We further showed that the SetCys peptide does not interfere with NCL even when the thioester component features a sterically demanding amino acid at its C terminus, typically a valine residue (Supplementary Figs. 62–65, Supplementary Table 2). We also verified that internal Cys residues are unable to activate SetCys residue, which is therefore useful for the production of Cys-rich peptides such as conotoxin OIVA (Supplementary Methods, Supplementary Fig. 5). Thus, the background NCL observed for a SetCys peptide in the presence of MPAA is unable to perturb a regular NCL involving a Cys peptide.

The most striking property of SetCys was observed when the SetCys peptide was subjected to the strong reducing conditions imposed by DTT or TCEP (Fig. 2a, property 4). In this case, the

reaction cleanly furnished the Cys peptide. We further documented that the reaction of a SetCys peptide with a peptide aryl or alkyl thioester in the presence of TCEP furnished a ligation product featuring a native Cys residue at the ligation junction (Fig. 2a, property 5). By contrast, the loss of the *N*-alkyl substituent was not observed when the sulfur analog of SetCys, featuring a 2-mercaptoethyl group on the α-nitrogen, was treated similarly, even after extended reaction times[24,25] (Fig. 2b, Supplementary Methods). The reactivity observed for SetCys depends specifically on the presence of selenium in its structure and, in that respect, SetCys allows to appreciate the high difference in reactivity that can exist between thiol and selenol compounds[26].

**Insights into the conversion of SetCys to a Cys residue.** From a mechanistic point of view, the loss of the selenoethyl group from the SetCys residue seems unlikely to involve radical intermediates since the reaction proceeds well in the presence of a large excess of sodium ascorbate and MPAA[27,28], two reagents known to be

powerful quenchers of alkylselenyl or alkylthiyl radicals. Omitting ascorbate during the treatment of SetCys peptide **1** by TCEP yields the deselenized peptide Et-CALKEPVHGV-NH$_2$ as the major product, whose formation competes against the loss of the selenoethyl arm (Supplementary Methods). Furthermore, the loss of the selenoethyl limb is also observed when dithiothreitol is used as a reducing agent, definitely ruling out the possibility that the reaction might involve a classical TCEP-induced dechalcogenation process[29,30]. Further insights into the species involved in the reaction come from the data shown in Fig. 3b, which presents the effect of pH on the rate of selenoethyl limb removal from model SetCys peptide **1** (Supplementary Table 3). The pH–rate profile of the conversion of reduced SetCys peptide **2** into cysteinyl peptide **3** shows a maximum at pH 6.0 ± 0.04 and two inflexion points at pH 4.8 and 7.3, which likely correspond to the pK$_a$s of the SetCys selenol and ammonium groups, respectively. These values are in agreement with the pK$_a$ values either reported for simple 2-selenoethylamines[31] and Cys derivatives[32] or estimated by calculation (Fig. 3c). The fact that the pH-rate profile of the reaction corresponds to the predominance zone for the

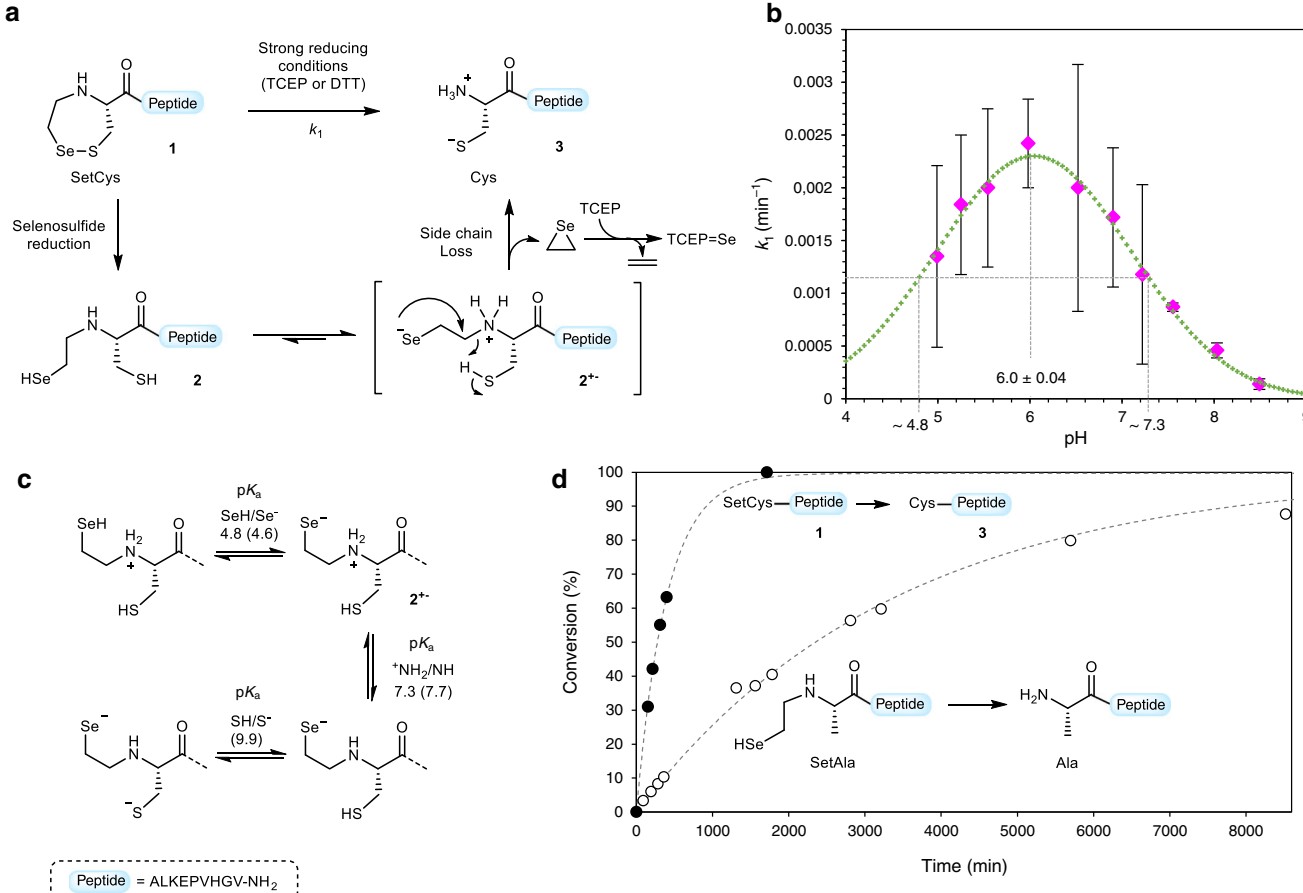

**Fig. 3 Insights into the conversion of SetCys to a Cys residue. a** Proposed mechanism for the loss of the *N*-selenoethyl appendage of N-terminal SetCys peptides. **b** Experimental pH–rate profile of the conversion of N-terminal SetCys peptide to the corresponding N-terminal cysteinyl peptide (magenta diamonds). Each of the 10 data points represents the value of the reaction rate $k_1$ obtained from different reactions conducted at the considered pH (1 mM SetCys peptide **1** in 0.1 M phosphate buffer, 6 M Gn·HCl, 200 mM MPAA, 100 mM TCEP, 100 mM sodium ascorbate, 37 °C). Data were fitted by chi$^2$ minimization. Associated standard errors (SE) were estimated based upon the covariance matrix resulting from nonlinear regression fitting (Supplementary Table 3). Data are presented as mean values ± SE. The data were then fitted to a Gaussian (green curve) to determine the pH values for the inflexion points (pH 4.8 and 7.3). **c** Different ionization states for the SetCys residue in open form. The numbers on the arrows correspond to the inflexion points determined in Fig. 3b and to pK$_a$ values calculated using ACDLabs® software (in parenthesis). **d** Rate of 2-selenoethyl limb cleavage in SetCys (filled circles, $k_1 = 2.42 \times 10^{-3}$ min$^{-1}$) and SetAla (open circles, $k_{SetAla} = 2.96 \times 10^{-4}$ min$^{-1}$) model peptides at pH 6. SetAla peptide was produced in situ from the corresponding diselenide by reduction in the presence of TCEP (1 mM SetCys or SetAla peptide in 0.1 M phosphate buffer, 6 M Gn·HCl, 200 mM MPAA, 100 mM TCEP, 100 mM sodium ascorbate, 37 °C).

selenoate/ammonium zwitterionic intermediate $2^{+-}$ led us to propose that the decomposition of SetCys proceeds through the intramolecular substitution of the ammonium group by the selenoate ion (Fig. 3a).

This mechanism results in the formation of an episelenide, which is known to be extremely unstable at room temperature and spontaneously decomposes into ethylene and selenium[18]. While selenium can be captured by TCEP in the form of the corresponding selenophosphine, whose formation was indeed observed in these reactions, detection of ethylene gas was made difficult by the small scale of synthesis.

The proposed mechanism is reminiscent of the cleavage of alkylamines by phenylselenol, albeit such reactions usually require elevated temperatures and/or assistance by metals[17,18,33]. Intrigued by the ease of SetCys to Cys conversion, we sought to determine if the SetCys thiol participates to the departure of the 2-selenoethyl limb. To this end, a N-(2-selenoethyl)-alanyl (SetAla) peptide analog was prepared and treated with MPAA/TCEP/ascorbate at the optimal pH for the SetCys to Cys conversion, i.e., pH 6.0 (Fig. 3d, Supplementary Methods). LC-MS analysis of the mixture showed the conversion of the SetAla residue into Ala, but at a rate considerably lower (~8.5-fold) than those measured for the SetCys to Cys conversion. This experiment shows that the departure of the 2-selenoethyl limb is greatly facilitated by the nearby SetCys thiol, perhaps by allowing an intramolecular proton transfer as depicted in Fig. 3a.

**Insights into the mechanism of SetCys-mediated ligation.** Having scrutinized the mechanism of SetCys conversion to a Cys residue under strong reductive conditions, we next examined the species involved during ligation with a peptide alkyl thioester under the same redox conditions. The monitoring of the reaction between SetCys peptide **1** and peptide thioester **4** indicated that a first ligation product **5**, containing an internal SetCys residue, accumulated within the first minutes and then slowly disappeared over 2 days in favor of peptide **6** featuring a native peptide bond to Cys (Fig. 4a–c).

Regarding the mechanism of SetCys-mediated ligation under strong reducing conditions, we hypothesized that the early formation of intermediate **5** is due to the interception of the reduced SetCys unit **2** by the thioester component (Fig. 4d). The latter is likely to be present in the form of the aryl thioester **7**, produced in situ from peptide alkyl thioester **4** by thiol-thioester exchange with the MPAA catalyst. Of the two nucleophilic sites present in reduced SetCys unit, the selenol moiety is probably the more reactive due to its lower $pK_a$ and higher nucleophilicity. The formation of tertiary amides of type **5** is known to be reversible in the conditions used for the ligation through their capacity to undergo an intramolecular nitrogen to selenium or sulfur acyl group migration[24,25,34]. Therefore, SetCys peptide **2** is constantly present in solution and escapes the SetCys/SetCys amide equilibrium by irreversibly losing its N-selenoethyl limb as discussed above. The Cys peptide **3** produced this way is expected to undergo a classical NCL reaction with aryl thioester **7** to yield ligated Cys peptide **6**. Although the proposed mechanism arises from the properties of the SetCys unit described in Fig. 2, we sought to confront it to kinetic data for validation. In addition, the model also tests the possibility of a direct conversion of SetCys amide **5** into final product **6**, being fully aware that the cleavage of the selenoethyl appendage from compound **5** through an ionic mechanism is unlikely due to the poor leaving group ability of imido nitrogens.

We first determined the rate constants associated (1) with the thiol-thioester exchange process involving peptide thioester **4** and MPAA ($k_{+2}$, $k_{-2}$), (2) with the conversion of SetCys peptide **2** into Cys peptide **3** ($k_1$), and (3) with the reaction of peptide aryl thioester **7** with Cys peptide **3** ($k_4$), from model reactions run separately (see Methods, Supplementary Tables 3 and 4). These rate constants were used to determine the remaining kinetic parameters $k_{+3}$, $k_{-3}$, and $k_5$ by fitting the experimental data of the SetCys-mediated ligation (circles and triangles in Fig. 4b) to the mechanistic model (Supplementaty Table 5). The quality of the fit (dashed lines in Fig. 4b) strongly suggests that the conversion of intermediate **5** to peptide **6** exclusively occurs through SetCys peptide **2** since the kinetic constant for the direct process **5** → **6** is at least 100-fold below the value measured for the decomposition of the SetCys into the Cys residue, i.e. **2** → **3**. The second piece of information provided by the model is that the SetCys peptide in its reduced form reacts almost as fast with the peptide thioester component as does the Cys peptide. Finally, the model tells us that the loss of the N-selenoethyl side chain from SetCys peptide **2** is the rate-limiting step of SetCys-mediated ligation.

**SetCys mediated one-pot multisegment assembly of proteins.** Having characterized the differential reactivity of the SetCys unit under mild and strong reducing conditions, we further sought to develop a simple process enabling the synthesis of cyclic proteins using SetCys as a redox switch (Fig. 5a, b). The motivation for this particular application comes from the observation that although a few studies pointed out the potential of protein cyclization for improving protein thermal stability[35,36], resistance to proteolytic degradation[36], and potency[37], this modification has not so far been widely used for the design of protein therapeutics. This situation contrasts with the success of small cyclic protein scaffolds such as cyclotides used as platforms for drug design[38], and the frequent use of macrocyclization in the development of small peptidic drugs[39]. The fact that protein cyclization is seldom used for protein optimization cannot be ascribed to inappropriate N–C distances, because half of the protein domains found in the protein data bank (PDB) have their C and N extremities joinable by linkers made of a few amino acids[40]. Rather, this situation reflects the paucity of tools for building cyclic proteins in a modular approach that facilitates the optimization of the linker joining the N- and C-termini[41].

Classical methods leading to cyclic proteins involve the macrocyclization of a bifunctional and linear precursor[42], primarily by using the NCL reaction[1] between a C-terminal thioester group and an N-terminal cysteine residue[2,43]. Following this strategy, the optimization of the linker requires the production of a library of extended precursors of varying length and composition, an approach that inevitably makes the production of cyclic analogs cumbersome.

In this work, we sought to develop a modular one-pot method enabling the grafting of the linker to a unique linear protein precursor (Fig. 5a, b). This can be achieved by exploiting the silent properties of the SetCys residue under mild reducing conditions for performing the first NCL (Fig. 2a, property 3), and then by using it as a redox switch for triggering the second cyclative NCL (Fig. 2a, property 5). Regarding the acyl donors, the process utilizes the good reactivity of alkyl thioesters in the presence of MPAA for the first ligation step. The second acyl donor is introduced as the *bis*(2-sulfenylethyl)amido (SEA)[44] latent thioester surrogate, which nicely complements the SetCys unit since it can also be activated in the presence of TCEP[45,46]. The Cys/SEA couple of functional groups is located on the linear protein precursor, while the SetCys/alkyl thioester functionalities are on the linker peptide. In the presence of MPAA alone, the Cys-mediated NCL with the peptide alkyl thioester (Step 1, Fig. 5b) exclusively yields a bifunctional polypeptide intermediate, which is activated at both ends by the addition of TCEP in one-pot (Step 2) to produce the backbone-cyclized polypeptide

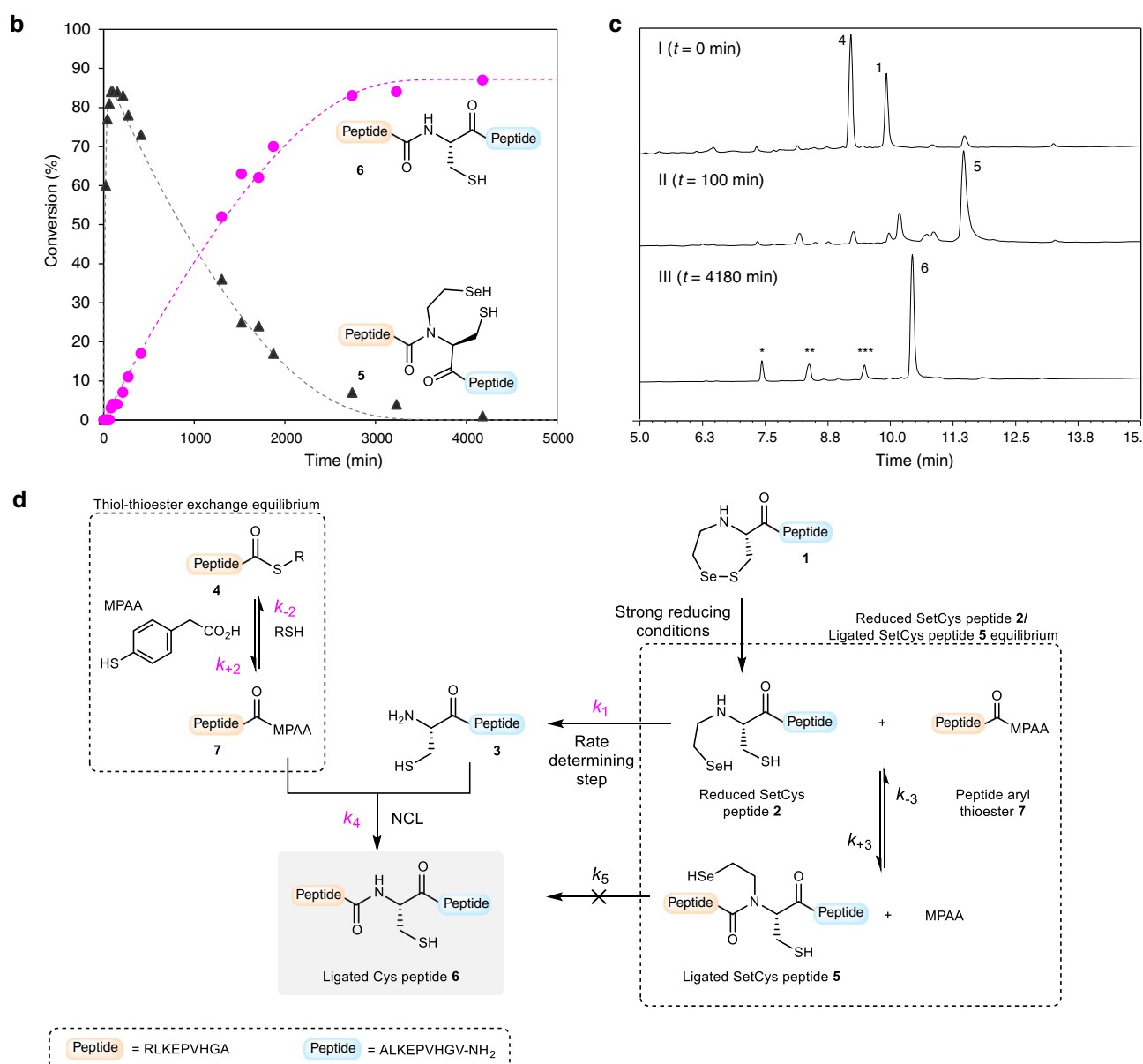

**Fig. 4 Insights into the mechanism of SetCys-mediated ligation. a** Ligation of SetCys peptide **1** with peptide thioester **4** in NCL standard conditions yields peptide **6** featuring a native peptide bond to cysteine. **b** RP-HPLC monitoring of the formation of peptides **5** (black triangles) and **6** (magenta dots) throughout the course of the reaction. Fitting curves for each compound are represented by dashed lines. Each aliquot corresponds to one data point on each trace and was measured once ($n = 1$). **c** Representative RP-HPLC chromatograms of the reaction in **a** at $t = 0$, 100, and 4180 min (the labeled peaks *, **, *** correspond to TCEP derived-selenophosphine, hydrolysis side-product of peptide thioester **4**, and Cys peptide **3**, respectively). **d** Proposed mechanistic model for the ligation of $N$-(2-selenoethyl)cysteine peptides under strong reducing conditions. The rate constants were obtained by software-assisted numerical integration of rate equations (Kintek explorer™). Rate constants and associated standard errors were estimated based upon the covariance matrix resulting from nonlinear regression fitting (Supplementary Table 5). Rate constants in magenta were determined separately from model reactions.

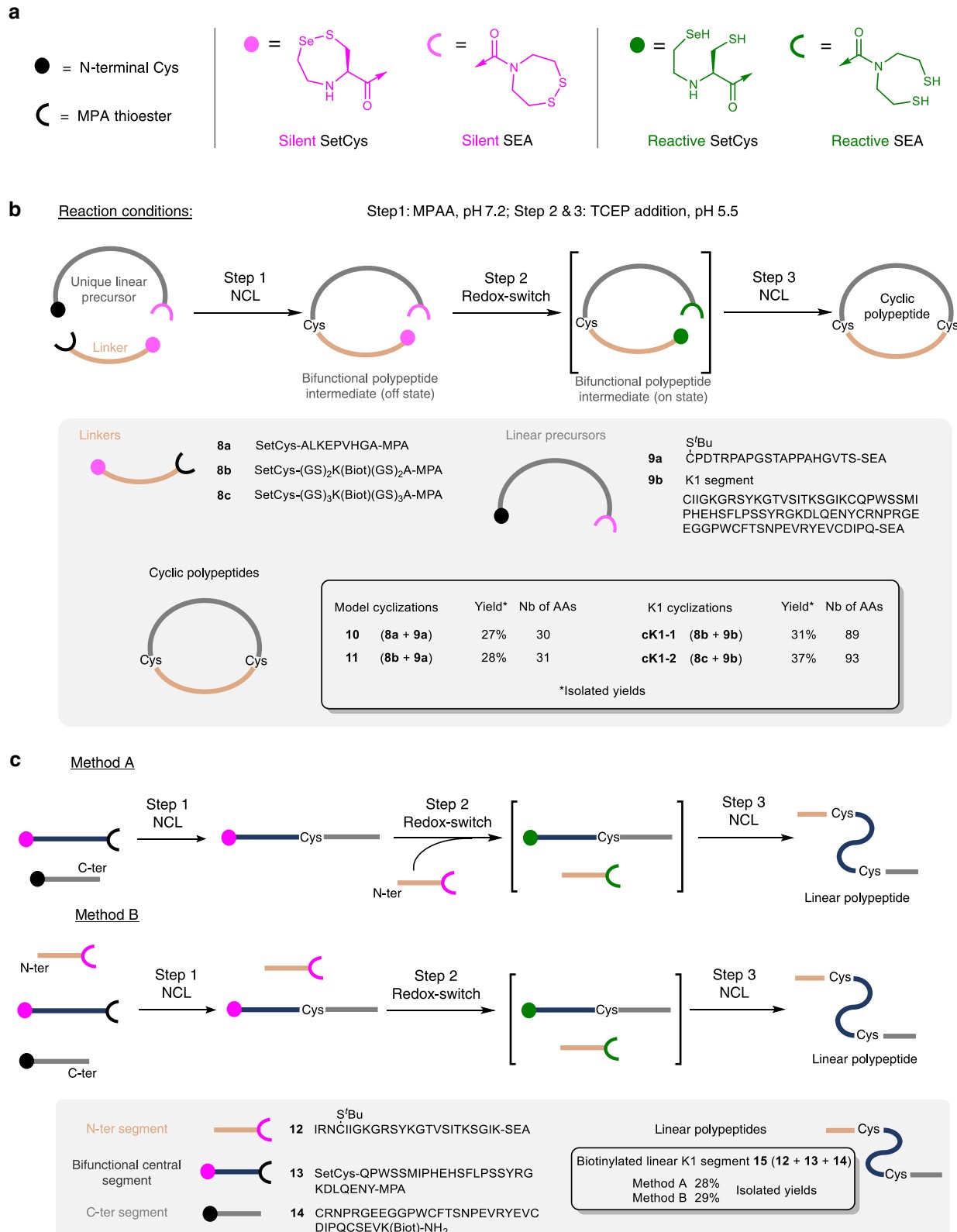

**Fig. 5 SetCys redox switch enables the straightforward synthesis of linear and cyclic proteins in one-pot. a** Schematic representations of peptidic N- and C-ter extremities. **b** One-pot grafting of a peptide linker to a unique linear precursor yields cyclic polypeptides. Application to the total synthesis of cyclic and biotinylated analogs of HGF/SF K1 domain (**cK1-1**, **cK1-2**). **c** One-pot synthesis of linear K1 domain.

(Step 3). The process is highly tolerant of polypeptide length as similar isolated yields were obtained for the production of medium to large cyclic peptides (30-93 amino acids, ~27–37% overall yields). The examples include the synthesis of two cyclic and biotinylated variants of HGF/scatter factor (HGF/SF) kringle 1 domain (K1), i.e., **cK1-1** and **cK1-2**, from a unique 78 residue linear precursor. These cyclic polypeptides differ by the length of the linker joining N- and C-termini of the K1 protein (10 and 14 residues, respectively).

SetCys chemistry proved equally useful for the C-to-N one-pot assembly of linear polypeptides from three peptide segments (Fig. 5a, c). The peptide segments could be added sequentially (Fig. 5c, Method A) or mixed altogether from the beginning of the assembly process (Fig. 5c, Method B) with equal selectivity and efficiency.

**Folding and bioactivity of biotinylated K1 cyclic analogs.** The signaling of the HGF/SF/MET system plays a crucial role in the regeneration of several tissues such as the liver or the skin, while its deregulation is often observed in cancer. The HGF/SF K1 domain contains the main HGF/SF binding site for the MET tyrosine kinase receptor and thus constitutes an interesting platform for designing future drugs based on this couple of proteins[47]. In this study, we sought to produce cyclic analogs of the K1 domain to investigate the tolerance of the K1/MET signaling system to this modification. The X-ray crystal structure of the K1 domain shows that its tertiary structure is made up of a series of loops stabilized by three disulfide bonds[48] (Fig. 6a). The N- and C-terminal cysteine residues are on the opposite side of the MET-binding loop and are linked by a disulfide bond. The N- and C-termini are thus close in space and can be joined by a peptide linker made of a few amino acid residues which include a biotinylated lysine residue. The latter is used to multimerize the ligand using streptavidin (**S**) due to the observation that multivalent presentation of the K1 domain is important for achieving high binding and agonistic activities[47].

The successful synthesis of the cyclic K1 polypeptides **cK1-1** and **cK1-2** set the stage for the folding step. **cK1-1** and **cK1-2** were folded into **cK1-1f** and **cK1-2f**, respectively, using the glutathione/glutathione disulfide redox system[46]. The folding mixtures converged to a major form after 24 h and were purified by dialysis (Fig. 6c). Extensive proteomic analysis of the folded proteins **cK1-1f** and **cK1-2f** showed the exclusive formation of the native pattern of disulfide bonds between Cys128–206, Cys149–189, and Cys177–201 as shown in Fig. 6b. Thus, the cyclization does not perturb the correct pairing of the Cys residues.

**cK1-1f** and **cK1-2f** proteins were first analyzed for their capacity to bind to the recombinant MET extracellular domain. The competitive AlphaScreen® assay with recombinant NK1 protein showed that the backbone-cyclized proteins **cK1-1f** and **cK1-2f** were ~10-fold less potent in binding the MET receptor than the biotinylated analog **K1B** (Fig. 6d). This result was unexpected because the cyclization site is diametrically opposite the MET binding site. The capacity of the cyclic K1 proteins to activate the MET receptor was further examined using human HeLa cells (Fig. 6e). MET phosphorylation induced by the cyclic K1 proteins was found to be less than that observed with the reference **K1B** analog. However, the tested K1 analogs triggered downstream signaling pathways, i.e., phosphorylation of AKT and ERK, with almost equal potency. Because previous studies showed marked differences between MET phosphorylation levels and the strength of MET-specific phenotypes induced by HGF or HGF mimics[49], we further analyzed the capacity of the different K1 proteins to trigger the scattering of human cells in vitro

(Fig. 6f). In this assay using human Capan 1 cells, the cyclic proteins behaved similarly to the reference protein **K1B** in the concentration range tested (10 pM–100 nM) by their capacity to induce a mesenchymal-like phenotype and cell scattering. To summarize, cyclization in this case results in a significant loss of affinity, although the backbone cyclization site is opposite to the receptor-binding site. However, this loss of affinity does not translate into the cell scattering activity. This work highlights the need for simple synthetic methods toward cyclic proteins to rapidly investigate the interest of backbone cyclization for improving protein properties.

## Discussion

The control of Cys reactivity is of paramount importance in chemical biology research owing to the frequent use of its thiol functionality for protein synthesis as discussed in this work, but also for the site-specific modification of peptides or proteins. The reactivity of the SetCys residue is peculiar in that the loss of the 2-selenoethyl arm through carbon–nitrogen bond cleavage proceeds under very mild conditions in water (Fig. 2a). This process which does not occur with the sulfur SutCys (Fig. 2b) analog illustrates once again the high difference in reactivity that can exist between selenium and sulfur compounds. However, in the case of SetCys, the reactivity of the selenol and thiol groups are intimately coupled as demonstrated by the reduced sensitivity of SetAla residue to carbon–nitrogen bond cleavage (Fig. 3d). Various water soluble and harmless reductants are available to the protein chemist so as to achieve a large array of reducing power and to place SetCys-mediated NCL reactions under redox control. In this sense, the use of SetCys as a precursor to Cys is conceptually different to existing methods and complements the chemical toolbox for protein chemical synthesis. Besides the ease of SetCys to Cys conversion, another hallmark of SetCys reactivity is certainly the reversibility of the peptide bond to SetCys under classical NCL conditions (Fig. 4). Taken together, these properties enable to diversify the type of peptide products that can be obtained through NCL such as peptides featuring an internal SetCys residue (Fig. 2a), or linear and cyclic proteins featuring native internal Cys residues (Fig. 5).

In conclusion, the cyclic selenosulfide derivative of cysteine, i.e. SetCys, shows an array of reactivities in water depending on the reducing power of the solution. A striking property of SetCys is its conversion to a native Cys residue by cleavage of a nitrogen–carbon bond under extremely mild conditions if a strong reducing agent is present in the solution. This transformation also occurs in the presence of a peptide thioester component and thus leads to the production of a native peptide bond to Cys by NCL. By contrast, SetCys remains silent during NCL if it is conducted under mild reducing conditions. The redox-switch properties of SetCys are particularly adapted for flexible synthetic designs that involve nothing more than common additive that are routinely used in NCL reaction.

## Methods

**Reagents.** 1-[Bis(dimethylamino)methylene]-1*H*-1,2,3-triazolo[4,5-*b*]pyridinium 3-oxid hexafluorophosphate (HATU), 2-(1*H*-benzotriazol-1-yl)-1,1,3,3-tetra-methyluronium fluorophosphate (HBTU) and *N*-Fmoc-protected amino acids were obtained from Iris Biotech GmbH. Piperidine, diisopropylcarbodiimide (DIC), *N*-hydroxybenzotriazole (HOBt), MPAA (97%, MPAA), 3-mercaptopropionic acid (MPA), tris(2-carboxyethyl)phosphine hydrochloride (≥98%, TCEP·HCl), thiophenol, triisopropylsilane (TIS), guanidine hydrochloride (≥99%, Gn·HCl), hydrochloric acid (reagent grade, 37%), and sodium hydroxide (pellets, 97%) as well as other reagents were purchased from Sigma Aldrich, Acros Organics or Merck and were of the purest grade available. Peptide synthesis grade *N*,*N*-dimethylformamide (DMF), dichloromethane, diethyl ether, acetonitrile, heptane, LC-MS-grade acetonitrile (0.1% TFA), LC-MS grade water (0.1% TFA), *N*,*N*-diisopropylethylamine (DIEA), acetic anhydride (Ac₂O), and trifluoroacetic acid (TFA) were purchased from Biosolve and Fisher-Chemical. NovaSyn TGR

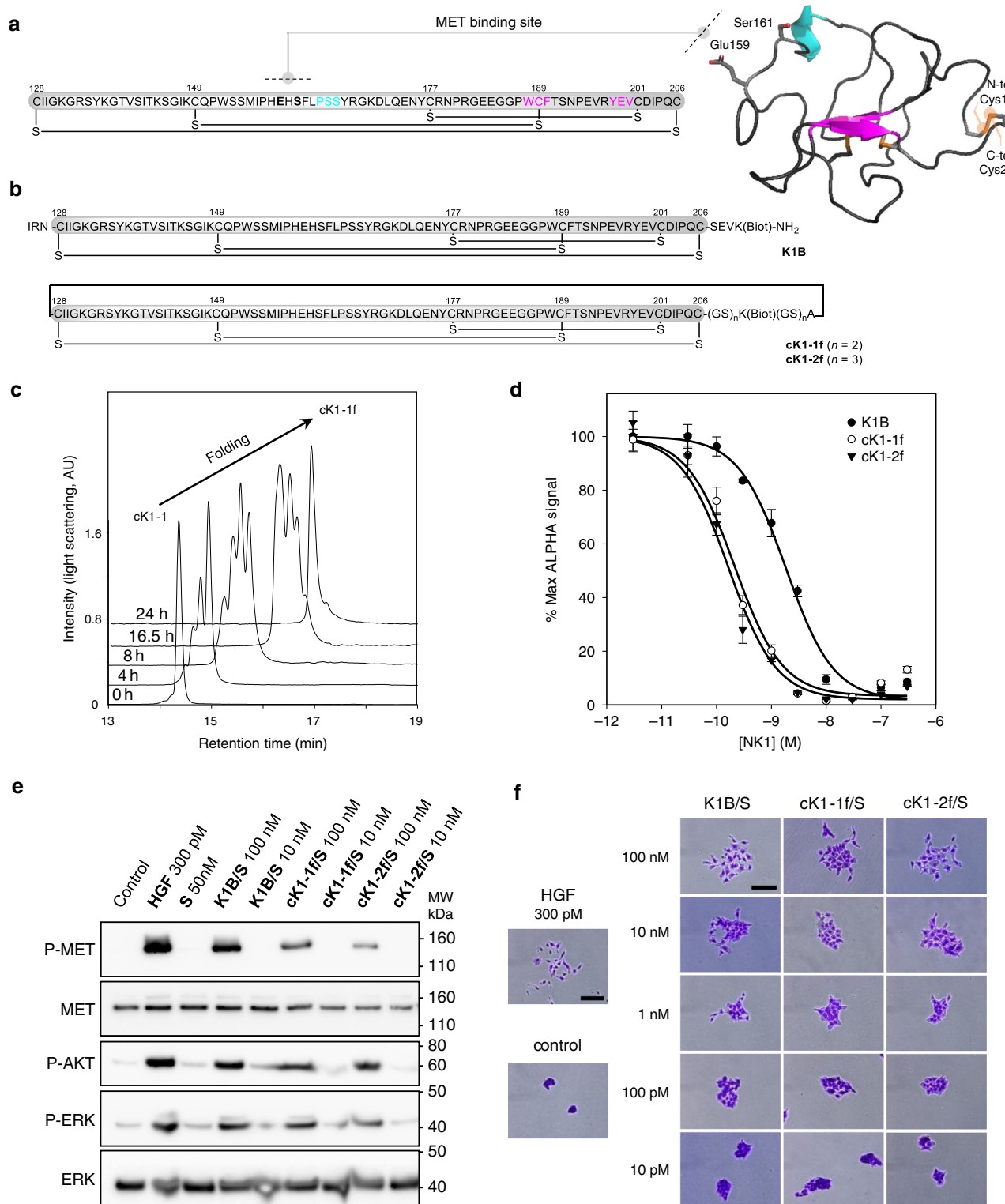

solid support was purchased from Sigma Aldrich. *Bis*(2-sulfenylethyl)amino trityl 1% divinylbenzene cross-linked polystyrene solid support (SEA PS) was produced on demand by Roowin. Solvents and reagents were used as received.

**Instrumentation.** The following instrumentation was used to purify and to characterize the amino acids and peptides synthesized in this work.

[1]H and [13]C NMR spectra were recorded on a Bruker Advance-300 spectrometer operating at 300 and 75 MHz, respectively. The spectra are reported as parts per million (ppm) down field shift using tetramethylsilane as internal references. The data are reported as chemical shift ($\delta$), multiplicity, relative integral, coupling constant (J Hz), and assignment where possible.

Analytical HPLC as well as micropreparative HPLC were performed with a Thermofisher system on a reverse-phase column XBridge BEH300 C18 (3.5 μm, 300 Å, 4.6 × 150 mm) using a linear gradient of increasing concentration of eluent B in eluent A (eluent A: 0.1% by vol. of TFA in water; eluent B: 0.1% vol. of TFA in acetonitrile or acetonitrile/water: 4/1 v/v). The column eluant was monitored by UV at 215 nm. Preparative HPLC Waters or Gilson systems equipped with a C18 reverse-phase column and a UV detector at 215 nm were used for the purification of peptides on larger scale.

Purified products as well as reaction mixtures were characterized by analytical LC-MS (Waters 2695 LC/ZQ 2000 quadripole) on a reverse-phase column XBridge BEH300 C18 (3.5 μm, 300 Å, 4.6 × 150 mm) using a linear gradient of increasing concentration of eluent B in eluent A. The column eluate was monitored by UV at

**Fig. 6 Folding and biological activity of biotinylated K1 cyclic analogs. a** Primary and tertiary structure of HGF/SF K1 domain (pdb entry 1BHT). **b** Biotinylated K1 analogs tested for their capacity to bind MET receptor and induce MET-specific phenotypes. The pattern of disulfide bonds determined experimentally corresponds to the native pattern found in K1 domain X-ray crystal structures. **c** LC-MS monitoring of the folding of **cK1-1** peptide into **cK1-1f**. **d** Competitive AlphaScreen® assay with recombinant NK1 protein. **K1B** or **cK1-1f** or **cK1-2f** were mixed with increasing concentrations of NK1 and with extracellular MET domain fused with human IgG1-Fc (MET-Fc) and incubated with streptavidin AlphaScreen® donor beads and Protein A acceptor beads. Data are presented as normalized percentage of maximal expected signal, i.e. without NK1 competition. Error bars represent the standard deviation (SD) of technical replicates ($n = 3$). **e** MET phosphorylation assay. HeLa cells were treated for 10 min with 300 pM mature HGF/SF (HGF), or with 10 nM/ 100 nM **K1/S**, **cK1-1f/S**, and **cK1-2f/S**. Cell lysates were then analyzed by specific total MET and ERK or phospho-MET, phospho-Akt and phospho-ERK western blot. Total MET and ERK were used as loading controls after membrane stripping and re-probing. This western blot is representative of two independent experiments ($n = 2$). **f** Cell scattering assay. Capan isolated cell islets were incubated for 18 h in culture media with 300 pM mature HGF/SF (HGF), or 100, 10, 1 nM and 100 and 10 pM **K1B**, **cK1-1f**, and **cK1-2f**. Cell scattering was observed after staining at ×40 (HGF and control, scale bar 500 μm) and ×200 magnification (cK1 treated, scale bar 100 μm). These micrographs are representative of two independent experiments ($n = 2$).

215 nm, evaporative light scattering (ELS, waters 2424) and electrospray ionization mass spectrometry (ESI-MS). Using similar eluants, analytical UPLC-MS analyses were performed on System Ultimate 3000 UPLC (Thermofisher) equipped with a column Acquity peptide BEH300 C18 (1.7 μm, 2.1 × 100 mm), a diode array detector, a charged aerosol detector (CAD), and a mass spectrometer (Ion trap LCQfleet).

MALDI-TOF mass spectra were recorded with a Bruker Autoflex Speed using alpha cyano 4-hydroxycinnaminic acid, sinapinic acid, or 2,5-dihydroxybenzoic acid (DHB) as matrix. For HRMS analyses, this work has benefited from the facilities and expertise of "Service HPLC-MASSE" (Institut de Chimie des Substances Naturelles, Centre de Recherche de Gif sur Yvette, CNRS, 91198, Gif sur Yvette, France).

Optical rotations were recorded at 20 °C on a Perkin-Elmer 343 digital polarimeter at 589 nm with a cell path length of 10 cm. The determination of optical purity for amino acid residues in peptide sequence was done via chiral GC-MS following total acid hydrolysis in deuterated aqueous acid (C.A.T. GmbH & Co. Chromatographie und Analysentechnik KG, Heerweg 10, D-72070 Tübingen, Germany).

The IR spectra were recorded on a Bruker FT-IR spectrometer ALPHA.

**Software**. Kintek Global Kinetic Explorer Version 8.0.190823 was used for kinetic modelizations.

**Amino acid purifications and characterizations**. The SetCys and SetAla amino acids were prepared using classical instrumentations and methods used in synthetic organic chemistry. The synthesized compound were purified by column chromatography and were characterized by [1]H and [13]C NMR, MS, HRMS, IR, and polarimetric analyses.

**Peptide purifications and characterizations**. The peptides were purified by reversed-phase HPLC and the fractions containing the target peptide were then combined, frozen, and lyophilized. Synthesized peptides were characterized by LC-MS and MALDI-TOF analyses. Sequence or disulfide bridge pattern of key peptides were confirmed using MALDI-TOF MS-MS analysis or proteomic analysis (enzymatic digestion followed by MS characterization of the fragments).

**Amino acid synthesis**. Fmoc-protected SetCys amino acid was synthesized from cysteine methyl ester hydrochloride in five steps with 55% overall yield, as described in the Supplementary Methods section and in Supplementary Fig. 1. In brief, the thiol function of the cysteine methyl ester hydrochloride was first protected with a trityl (Trt) group and the selenoethyl arm which is *Se*-protected with a *para*-methoxybenzyl (PMB) group was then inserted by reductive amination reaction[50]. After Fmoc protection of the secondary amine, the cyclic backbone of the SetCys residue was formed by oxidation with iodine, through a process that induced in one step removal of the PMB and Trt protecting groups and the formation of the selenosulfide bridge. Finally, the deprotection of the carboxylic acid function in acidic conditions provided the target Fmoc-protected SetCys amino acid that was subsequently used in the synthesis of SetCys peptides by standard Fmoc solid phase peptide synthesis methods.

Fmoc- and PMB-protected SetAla amino acid was synthesized from alanine methyl ester hydrochloride in three steps with 30% overall yield, as described in the Supplementary Methods section and in Supplementary Fig. 2. The *N*-selenoethyl arm of the target amino acid, *Se*-protected with a PMB group, was inserted by reductive amination reaction[50]. After Fmoc-protection of the secondary amine, deprotection of the carboxylic acid function in acidic conditions provided the expected Fmoc- and PMB-protected SetAla amino acid that was subsequently used in the elongation of SetAla peptides by standard Fmoc solid phase peptide synthesis methods.

**Peptide synthesis**. Peptides **1**, **4**, **8a–c**, **9a**, and **12–14** and all the segments needed to produce K1 analog **9b** were synthesized using standard Fmoc solid phase peptide synthesis methods. Peptide amides were prepared on a NovaSyn TGR solid support (0.25 mmol g[−1], Supplementary Fig. 3), whereas the sequences of the SEA and MPA thioesters peptides were assembled on SEA PS solid support (0.15 mmol g[−1], Supplementary Fig. 4).

*Automated peptide elongation*: Unless otherwise stated, peptide elongation was performed at room temperture (rt) using an automated peptide synthesizer. In brief, amino acids (10 equiv) were activated using HBTU (9.5 equiv)/DIEA (10 equiv) in DMF. The peptidyl solid support was acetylated after each coupling step using Ac₂O/DIEA/DMF 10/5/85 v/v/v. The removal of the Fmoc group was performed by treating the peptidyl solid support with DMF/piperidine 80/20 v/v.

*Coupling of the first amino acid to SEA PS solid support*: The first amino acid (10 equiv) was manually coupled to SEA PS solid support (1 equiv) at rt using HATU (10 equiv)/DIEA (20 equiv) activation in DMF. The absence of unreacted secondary amino groups was tested using chloranil colorimetric assay. A capping step was then performed using Ac₂O/DIEA/DMF 10/5/85 v/v/v. Finally, the solid support was treated with DMF/piperidine 80/20 v/v to remove the Fmoc protecting group of the coupled residue.

*Coupling of Fmoc-Lys(biotin)-OH*: To a solution of HATU (190 mg, 0.500 mmol, 5 equiv) in DMF (3 mL) were successively added Fmoc-Lys(biotin)-OH (300 mg, 0.504 mmol, 5 equiv) and DIEA (173 μL, 1.00 mmol, 10 equiv) and the mixture was stirred at rt for 2 min. The preactivated amino acid was then added to the solid support (0.1 mmol, 1 equiv) swelled in the minimal volume of DMF and the beads were agitated at rt for 1.5 h. A capping step was then performed using Ac₂O/DIEA/ DMF 10/5/85 v/v/v. Finally, the peptidyl solid support was treated with DMF/ piperidine 80/20 v/v to remove the Fmoc-protecting group of the Lys(biotin) residue.

*Coupling of the Fmoc-SetCys-OH amino acid*: To a solution of Fmoc-SetCys-OH (44.8 mg, 0.100 mmol, 2 equiv) in DMF (2.5 mL) were successively added HOBt·H₂O (15.3 mg, 0.100 mmol, 2 equiv) and DIC (15.6 μL, 0.100 mmol, 2 equiv) and the mixture was stirred at rt for 2 min. The preactivated amino acid was then added to the solid support (0.05 mmol, 1 equiv) swelled in the minimal volume of DMF and the beads were agitated at rt for 3 h. A capping step was then performed using Ac₂O/DIEA/DMF 10/5/85 v/v/v. Finally, the peptidyl solid support was treated with DMF/piperidine 80/20 v/v to remove the Fmoc-protecting group of the SetCys residue.

*Final peptide deprotection and cleavage*: At the end of the elongation process, the solid support was washed with CH₂Cl₂ (3×), shrinked with diethyl ether (3×), and dried in vacuo. The elongated peptide was cleaved from the solid support using TFA with appropriate scavengers as indicated in each case, precipitated by addition of cold diethyl ether/heptane 1/1 v/v (20 mL per mL of TFA cocktail), and recovered by centrifugation. The solid was washed with cold diethyl ether/heptane 1/1 v/v (40 mL), dissolved in water and the solution was lyophilized. If needed, crude peptide amides were purified by reversed-phase HPLC. In the case SEAon peptides, crude peptides were directly used in the next step (oxidation of the SEA group, transthioesterification with MPA) without further purifications.

*Oxidation of the SEA group (SEA^on → SEA^off)*: The SEA^on peptide was dissolved in AcOH/water 1/4 v/v. A solution of iodine in DMSO (100 mg mL⁻¹) was added dropwise until persistence of the yellow color of iodine in the reaction mixture, indicating the complete oxidation of the SEA group. After 30 s of stirring, a solution of DTT in AcOH/water 1/4 v/v (≈100 mg mL⁻¹) was added to consume the excess of iodine. Then the peptide was immediately purified by reversed-phase HPLC (see ref. [45] for detailed experimental procedures).

*Transthioesterification (SEA^on → MPA)*: SEA peptides can be converted into thioesters derived from 3-mercaptopropionic acid (MPA) by reaction with an excess of MPA at pH 4.0 (ref. [45]). This reaction is usually carried out in the presence of TCEP starting from SEA^on or SEA^off peptides. However, the presence of TCEP is known to induce the conversion of SetCys into Cys. Consequently, a TCEP-free method for the synthesis of MPA thioesters featuring an N-terminal SetCys residue was developed. Such an experimental approach is illustrated with the synthesis of peptide **8a** (without Gn·HCl additive) and SetCys-K1[150-176]- MPA peptide **13** (with Gn·HCl additive).

**Synthesis of peptide *8a*:** SetCys-ALKEPVHGA-SEA[on] peptide was synthesized on 0.08 mmol scale as described in the general procedure. TFA/H$_2$O/TIS/thiophenol 92.5/2.5/2.5/2.5 v/v/v/v (8 mL) was used as the cleavage cocktail. The SEA[on] peptide which was recovered by precipitation from Et$_2$O/heptane was dissolved immediately in a solution of MPA (1 mL) in water (19 mL) at pH 4.0. After 16 h stirring under inert atmosphere at 37 °C, the reaction mixture was acidified with 10% aqueous AcOH (10 mL) and extracted with Et$_2$O (5×) to remove the excess of MPA. Purification of the crude by semi-preparative HPLC (eluent A: 0.1% TFA in water, eluent B: 0.1% TFA in CH$_3$CN/water 4/1 v/v, 6 mL min$^{-1}$, 0–35% eluent B in 25 min, C18 column, detection at 215 nm, 30 °C) furnished SetCys-ALKEPVHGA-MPA peptide *8a* as a white solid after lyophilization (25.3 mg, 20%). Characterization of peptide *8a* by LC-MS is provided in the Supplementary Methods section.

**Synthesis of SetCys-K1[150-176]-MPA peptide *13*:** SetCys-K1[150-176]-SEA[on] peptide was synthesized on 0.05 mmol scale as described in the general procedure. TFA/H$_2$O/TIS/thiophenol/thioanisol 87.5/2.5/5/2.5/2.5 v/v/v/v/v (8 mL) was used as the cleavage cocktail. The SEA[on] peptide which was recovered by precipitation from Et$_2$O/heptane was added immediately in a solution of MPA (0.625 mL) in water (11.9 mL) at pH 4. Due to the low solubility of the SEA[on] peptide in the solution, Gn·HCl (8.58 g) was added until complete dissolution of the solid. The pH was then readjusted to 4.0 by addition of 6 M NaOH and the reaction mixture was stirred under inert atmosphere at 37 °C for 16 h. The reaction mixture was then acidified with AcOH (1.50 mL) and extracted with Et$_2$O (5×) to remove the excess of MPA. Purification of the crude by semi-preparative HPLC (eluent A: 0.1% TFA in water, eluent B: 0.1% TFA in CH$_3$CN, 20 mL min$^{-1}$, 0–10% eluent B in 5 min then 10–35% in 30 min, C18 column, detection at 215 nm, 30 °C) furnished SetCys-K1[150–176]-MPA peptide *13* as a white solid after lyophilization (34.7 mg, 17%). Characterization of peptide *13* by LC-MS is provided in the Supplementary Methods section.

**SetCys reactivity studies (Fig. 2a).** *Property 1.* SetCys peptide *1* (8 mM) is stable at pH 7.2 in the presence of 200 mM MPAA (Supplementary Fig. 54).

*Property 2.* The following experimental procedure that describes the preparation of the SetCys amide peptide RLKEPVHGA-SetCys-ALKEPVHGV-NH$_2$ by ligation of peptides *1* and *4* (Supplementary Fig. 55) was used to illustrate the property 2 of SetCys peptide *1*. To a solution of Gn·HCl (1.03 g) in 0.1 M, pH 7.4 phosphate buffer (1.08 mL) was added MPAA (60.6 mg) and the pH of the mixture was adjusted to 7.20 by addition of 6 M NaOH. SetCys peptide *1* (23.2 mg, 15.5 μmol, 1.2 equiv) and peptide thioester *4* (20.0 mg, 12.9 μmol, 1 equiv) were then successively dissolved in this solution (1.61 mL) and the reaction mixture was stirred at 37 °C for 24 h. The reaction was monitored by LC-MS (Supplementary Fig. 56) and the kinetic parameters were determined by numerical fitting using Kintek Global Explorer Software (Supplementary Figs. 57 and 58). After completion of the reaction, the mixture was then acidified with 10% AcOH in water (22.5 mL) and extracted with Et$_2$O (5×) to remove the MPAA. Purification of the crude by HPLC (eluent A: 0.1% TFA in water, eluent B: 0.1% TFA in acetonitrile/water 4/1 v/v, 50 °C, detection at 215 nm, 3 mL min$^{-1}$, 0–10% eluent B in 5 min, then 10–35% eluent B in 35 min, then 35–50% eluent B in 15 min, C18 column) provided RLKEPVHGA-SetCys-ALKEPVHGV-NH$_2$ as a white solid after lyophilization (15.1 mg, 41%). Full characterizations of the SetCys amide peptide by LC-MS and MALDI-TOF MS are provided in the Supplementary Methods section.

*Property 3.* The following experimental procedure that enables the comparison of Cys and SetCys peptide reactivity towards peptide thioester (Supplementary Fig. 62) was used to illustrate the property 3 of SetCys peptide *1*. To a solution of Gn·HCl (287 mg) in 0.1 M, pH 7.4 phosphate buffer (300 μL) was added MPAA (16.8 mg) and the pH of the mixture was adjusted to 7.2 by addition of 6 M NaOH. SetCys peptide *1* (1.32 mg, 0.881 μmol, 1.2 equiv), cysteinyl peptide CILKEPVHGV-NH$_2$ (1.27 mg, 0.884 μmol, 1.2 equiv) and peptide thioester *4* (1.14 mg, 0.735 μmol, 1 equiv) were then successively dissolved in this solution (92 μL) and the reaction mixture was stirred at 37 °C. Note that the reaction was repeated using the more hindered peptide thioester ILKEPVHGV-MPA featuring a C-terminal Val residue.

Whatever the acyl donor (*4* or ILKEPVHGV-MPA), the SetCys peptide *1* was recovered unreacted at the end of the ligation (Supplementary Figs. 63 and 64). In each case, the kinetically favored ligation (respectively *4* + CILKEPVHGV-NH$_2$ and ILKEPVHGV-MPA + CILKEPVHGV-NH$_2$) was monitored by HPLC and the kinetic parameters were determined by numerical fitting. The results are given in Supplementary Fig. 65 and Supplementary Table 2.

We also verified that SetCys residue is not reduced and activated by internal Cys residues by assemblying the Cys-rich conotoxin OIVA (CCGVONAACHOCVC KNTC-NH$_2$, O = hydroxyproline). As described in Supplementary Fig. 5 and Supplementary Methods, the linear peptide corresponding to conotoxin OIVA was synthesized from two peptide segments by temporally masking the N-terminal Cys amino acid of the target conotoxin peptide with a SetCys residue.

*Property 4.* The following procedure was used for studying the conversion of SetCys peptide *1* into Cys peptide *3* under strong reducing conditions. Illustrated with MPAA + TCEP as a reducing agent, such a procedure was also used to characterize the decomposition of SetCys residue in the presence of MPAA + DTT. To a solution of Gn·HCl (287 mg) in 0.1 M, pH 7.4 phosphate buffer (300 μL) were added TCEP·HCl (14.3 mg), sodium ascorbate (9.9 mg), and MPAA (16.8 mg) and

the pH of the mixture was adjusted to 7.2 by addition of 6 M NaOH. SetCys peptide *1* (0.96 mg, 0.64 μmol) was then dissolved in this solution (80 μL) and the reaction mixture was stirred at 37 °C. The conversion of SetCys into Cys was monitored by HPLC. For each point, a 2 μL aliquot was taken from the reaction mixture and quenched by adding 10% AcOH in water (100 μL). The sample was then extracted with Et$_2$O to remove MPAA prior to HPLC analysis.

HPLC monitoring of the reactions using respectively MPAA + TCEP and MPAA + DTT as a reducing agent are shown in Supplementary Figs. 66 and 67 and the kinetic parameters were determined by numerical fitting using Kintek Global Explorer Software (Supplementary Fig. 68 and Supplementary Table 1). Note that the description of a control experiment performed in the presence of TCEP but in the absence of ascorbate is provided in the Supplementary Methods and highlighted a strong deselenization side reaction in these conditions.

*Property 5.* The following experimental procedure that describes the preparation of the native peptide *6* by ligation of peptides *1* and *4* (Supplementary Fig. 75) was used to illustrate the property 5 of SetCys peptide *1*. To a solution of Gn·HCl (574 mg) in 0.1 M, pH 7.4 phosphate buffer (600 μL) were added TCEP·HCl (28.7 mg), sodium ascorbate (19.8 mg), and MPAA (33.6 mg) and the pH of the mixture was adjusted to 7.17 by addition of 6 M NaOH. Peptide thioester *4* (6.04 mg, 3.89 μmol, 1 equiv) and SetCys peptide *1* (7.00 mg, 4.67 μmol, 1.2 equiv) were then successively dissolved in this solution (212 μL) and the reaction mixture was stirred at 37 °C for 4 days. The progress of the ligation was monitored by HPLC and the kinetic data are provided in Supplementary Fig. 76. After completion of the reaction, the mixture was then acidified with 5% AcOH in water (10 mL) and extracted with Et$_2$O (5×) to remove the MPAA. Purification of the crude by HPLC (eluent A 0.1% TFA in water, eluent B 0.1% TFA in acetonitrile/water 4/1 v/v, 50 °C, detection at 215 nm, 3 mL min$^{-1}$, 0–10% eluent B in 5 min, then 10–25% eluent B in 30 min, C18 column) furnished peptide *6* as a white solid after lyophilization (6.81 mg, 64%). Full characterizations of peptide *6* are provided in the Supplementary Methods.

The description of the reactions using DTT/MPAA as a reducing agent or a peptide aryl thioester derived from MPAA in the absence of exogenous MPAA can also be found in the Supplementary Methods.

**Mechanistic studies (Fig. 3).** The general procedure for studying the effect of the pH on the rate of SetCys to Cys conversion is illustrated with the following conditions: 1 mM peptide, 200 mM MPAA, 100 mM TCEP, 100 mM ascorbate, pH 6.90 in 0.1 M phosphate buffer. To a solution of Gn·HCl (287 mg) in 0.1 M pH 7.4 phosphate buffer (300 μL) were added TCEP·HCl (14.3 mg), sodium ascorbate (9.9 mg), and MPAA (16.8 mg) and the pH of the mixture was adjusted to 6.90 by addition of 6 M NaOH. SetCys peptide *1* (0.48 mg, 0.32 μmol) was then dissolved in the buffered solution (320 μL) and the reaction mixture was stirred at 37 °C. The progress of the reaction was monitored by HPLC. For each point, an 8 μL aliquot was taken from the reaction mixture and quenched by adding 10% AcOH in water (100 μL). The sample was then extracted with Et$_2$O to remove MPAA prior to HPLC analysis. The kinetic data are shown in Supplementary Fig. 95.

The reactions leading to the loss of the SetCys *N*-selenoethyl appendage were performed once at each pH value from 4.99 to 8.49. Corresponding rate constants and associated standard errors (SE) were determined by chi$^2$ minimization fitting. Standard errors produced by nonlinear regression fitting were estimated based upon the covariance matrix using Kintek software. All the data are given in Supplementary Table 3.

The contribution of the Cys side-chain thiol to the rate of SetCys to Cys conversion was probed using SetAla peptide as described in Fig. 3d (Supplementary Methods).

**Kinetic model of SetCys-mediated NCL under strong reducing conditions (Fig. 4).** In order to test the model proposed in Fig. 4d and determine the different rate constants involved in it, the mechanistic study was first decomposed into simpler reactions, wherefrom kinetic parameters were extracted.

*Rate constants for a classical NCL ($k_{+2}$, $k_{-2}$, and $k_4$, Fig. 4d):* The kinetic profile of a model NCL reaction was first studied in order to determine the rate constants of the thiol-thioester exchange step ($k_{+2}$ and $k_{-2}$) and of the capture of the transient aryl thioester by a cysteinyl peptide ($k_4$) (Supplementary Fig. 102). At the peptide millimolar concentration range, NCL reactions are under the kinetic control of thiol–thioester exchanges. Such conditions do not allow to observe the accumulation of the transient aryl thioester and make the determination of $k_4$ rate constant impossible. In order to extract second-order rate constants, the model thioester and the Cys peptides were reacted at a respective concentration of 0.1 and 0.15 mM (in phosphate buffer 0.1 M, 6 M Gn·HCl, 200 mM MPAA, 100 mM TCEP, pH 7.2, 37 °C). Fifty microliters of the reaction mixture were collected every ~20 min and added to 50 μL of a 50% acetic acid aqueous solution. The sample was extracted by Et$_2$O (3×) and the aqueous layer was analyzed by RP-HPLC (eluent A 0.1% TFA in water, eluent B 0.1% TFA in CH$_3$CN/water 4/1 v/v, C18Xbridge BEH300 Å 5 μm (4.6 × 250 mm) column, gradient 0–50% B in 15 min, 1 mL min$^{-1}$, 30 °C, detection at 215 nm). The chromatograms were processed on the basis of the absorbance signal at 215 nm. All datasets were fitted with a three-exponential analytical function to obtain an estimate of the standard deviation. A subsequent numerical fit allowed determining rate constants $k_{+2}$, $k_{-2}$, $k_4$ (Supplementary Table 4, Supplementary Fig. 103). Reaction was performed once and associated

standard errors produced by nonlinear regression fitting were estimated based upon the covariance matrix using KinteK Explorer Software[TM].

*Rate constant for the loss of the N-selenoethyl appendage ($k_1$, Fig. 4d)*: The experimental approach for this reaction is documented in full above in the Methods section (section Mechanistic studies, Fig. 3). Note that the reduction of SetCys peptide **1** by TCEP was neglected since this reaction proceeds quickly at pH 7.2. As previously determined (Supplementary Table 3), $k_1 = 0.00118 \pm 0.00085$ min$^{-1}$ at pH 7.2, 37 °C.

*Kinetic model for SetCys-mediated NCL (Fig. 4b, d)*: The rate constants determined in the model NCL and 2-selenoethyl limb removal process from SetCys peptide **1** (i.e. $k_{+2}$, $k_{-2}$, $k_1$, $k_4$) were used to fit the kinetic data collected for the ligation of SetCys peptide **1** (Fig. 4b) with the model proposed in Fig. 4d. This procedure enabled the determination of $k_{+3}$, $k_{-3}$, and $k_5$ (Supplementary Table 5).

**One-pot assembly of cyclic polypeptides (Fig. 5a, b)**. The synthesis of **cK1-1** was used to illustrate the method leading to cyclic peptides (Supplementary Fig. 6). Note that the linear precursor **9b** of the K1 domain was assembled as described in Supplementary Methods and Supplementary Fig. 7. To a solution of Gn·HCl (287 mg) in 0.1 M, pH 7.4 phosphate buffer (300 μL) was added MPAA (16.8 mg) and the pH of the mixture was adjusted to 7.25 by addition of 6 M NaOH. Peptide **9b** (5.03 mg, 0.481 μmol, 1 equiv) and peptide **8b** (0.688 mg, 0.481 μmol, 1 equiv) were successively dissolved in the MPAA solution (120 μL) and the mixture was stirred at 37 °C. The progress of the ligation leading to the bifunctional peptide intermediate was followed by HPLC (Supplementary Fig. 113a).

After completion of the NCL ligation (2 h), the second step of the process leading to the cyclization of the peptide was induced by addition of TCEP. To a solution of Gn·HCl (1.72 g) in 0.1 M, pH 7.4 phosphate buffer (1.80 mL) were added TCEP·HCl (90.2 mg), sodium ascorbate (63.0 mg) and MPAA (100 mg) and the pH of the mixture was adjusted to 5.51 by addition of 6 M NaOH. The previous ligation mixture containing the bifunctional peptide intermediate was then diluted with the solution of TCEP (2.28 mL) and the mixture was stirred at 37 °C for 40 h. After completion of the reaction (Supplementary Fig. 113b), the mixture was acidified with AcOH (0.30 mL) and extracted with Et$_2$O (5×) to remove the MPAA. The crude was further diluted with water (9 mL) and purified by HPLC (eluent A 0.1% TFA in water, eluent B 0.1% TFA in acetonitrile/water 4/1 v/v, 50 °C, detection at 215 nm, 6 mL min$^{-1}$, 0–20% eluent B in 5 min, then 20–45% eluent B in 60 min, C18XBridge column) to give the cyclic peptide **cK1-1** as a white solid after lyophilization (1.71 mg, 31%). Characterizations of **cK1-1** by LC-MS and proteomic analysis are provided in the Supplementary Methods.

*Assembly of linear K1 polypeptide 15 (Fig. 5a, c), Method A*. Method A relies on the sequential addition of the peptide segments. As described in Supplementary Fig. 8, elongation of the SetCys peptide segment is first performed by NCL between segments **13** and **14** under the mild reductive conditions imposed by the presence of the MPAA catalyst. After completion of the NCL, the SEA peptide segment **12** is added to the reaction mixture and the second ligation leading to the target linear K1 polypeptide **15** is triggered by the concomitant activation of SetCys and the SEA groups by TCEP reduction.

To a solution of Gn·HCl (287 mg) in 0.1 M, pH 7.4 phosphate buffer (300 μL) was added MPAA (16.8 mg) and the pH of the mixture was adjusted to 7.22 by addition of 6 M NaOH. Peptide K1[177–209]-K(Biot)-NH$_2$ **14** (2.78 mg, 0.609 μmol, 1 equiv) and peptide SetCys-K1[150–176]-MPA **13** (2.50 mg, 0.610 μmol, 1 equiv) were successively dissolved in the MPAA solution (152 μL) and the mixture was stirred at 37 °C. The progress of the ligation leading to the elongated SeCys peptide was followed by UPLC-MS (Supplementary Fig. 121a).

After completion of the NCL ligation (2 h), the second step of the process that finalizes the assembly of the linear peptide K1 was performed by adding the SEA peptide K1[125–148]-SEA **12** and TCEP to the reaction mixture. To a solution of Gn·HCl (287 mg) in 0.1 M, pH 7.4 phosphate buffer (300 μL) were added TCEP·HCl (28.6 mg), sodium ascorbate (19.8 mg) and MPAA (16.8 mg) and the pH of the mixture was adjusted to 5.23 by addition of 6 M NaOH. SEA peptide K1[125-148]-SEA **12** (2.18 mg, 0.608 μmol, 1 equiv) was dissolved in the previous ligation mixture containing the elongated SetCys peptide and the resulting mixture was then diluted with the solution of TCEP (152 μL). The pH was readjusted to 5.52 by addition of 6 M HCl and the mixture was stirred at 37 °C for 48 h. After completion of the reaction (Supplementary Fig. 121b), the mixture was diluted with 7.5% AcOH in water (8 mL) and extracted with Et$_2$O (5×) to remove the MPAA. Purification of the crude by HPLC (eluent A: 0.1% TFA in water, eluent B: 0.1% TFA in acetonitrile, 6 mL min$^{-1}$, 0–10% eluent B in 5 min then 10–35% eluent B in 30 min, C18 column, rt, detection at 215 nm) provided the linear K1 polypeptide **15** as a white solid after lyophilization (1.96 mg, 28%).

*Assembly of linear K1 polypeptide 15 (Fig. 5a, c), Method B*. Method B relies on the latent properties of the SEA group, which enables the SEA peptide segment to be introduced at the beginning of the one-pot process, in mixture with the other reactive peptide segments. As described in Supplementary Fig. 9, a first NCL between segments **13** and **14** selectively produces the elongated SetCys peptide, the SEA peptide segment **12** being unreactive under the mild reductive conditions imposed by the MPAA catalyst. The SEA and the elongated SetCys peptide segments were then concomitantly activated by addition of TCEP to produce the final linear K1 polypeptide **15**.

To a solution of Gn·HCl (287 mg) in 0.1 M, pH 7.4 phosphate buffer (300 μL) was added MPAA (16.8 mg) and the pH of the mixture was adjusted to 7.28 by addition of 6 M NaOH. Peptide K1[125–148]-SEA **12** (2.18 mg, 0.608 μmol, 1 equiv), peptide K1[177–209]-K(Biot)-NH$_2$ **14** (2.78 mg, 0.609 μmol, 1 equiv), and peptide SetCys-K1[150-176]-MPA **13** (2.50 mg, 0.610 μmol, 1 equiv) were successively dissolved in the MPAA solution (152 μL) and the mixture was stirred at 37 °C. The progress of the ligation leading to the elongated SetCys peptide was followed by UPLC-MS (Supplementary Fig. 122a), the SEA peptide **12** being unreactive in these mild reductive conditions.

After completion of the NCL ligation (4.5 h), the second step of the process that ligates the elongated SetCys peptide with the SEA segment was induced by addition of TCEP. To a solution of Gn·HCl (287 mg) in 0.1 M, pH 7.4 phosphate buffer (300 μL) were added TCEP·HCl (28.6 mg), sodium ascorbate (19.8 mg), and MPAA (16.8 mg) and the pH of the mixture was adjusted to 5.40 by addition of 6 M NaOH. The previous ligation mixture containing the elongated SetCys peptide and the SEA segment **12** was then diluted with the solution of TCEP (152 μL). The pH was readjusted to 5.45 by addition of 6 M HCl and the mixture was stirred at 37 °C for 40 h. After completion of the reaction (Supplementary Fig. 122b), the mixture was diluted with 7.5% AcOH in water (8 mL) and extracted with Et$_2$O (5×) to remove the MPAA. Purification of the crude by HPLC (eluent A: 0.1% TFA in water, eluent B: 0.1% TFA in acetonitrile, 6 mL min$^{-1}$, 0–10% eluent B in 5 min then 10–35% eluent B in 30 min, C18 column, rt, detection at 215 nm) provided the linear K1 polypeptide **15** as a white solid after lyophilization (2.05 mg, 29%). Characterizations of the linear K1 peptide **15** by LC-MS, MALDI-TOF, and proteomic analysis are provided in the Supplementary Methods.

*Folding of cyclic cK1 polypeptides*. **cK1-1** and **cK1-2** were folded using the following procedure. The cyclic polypeptides **cK1-1** and **cK1-2** (33 nM/μmol final peptide concentration) were dissolved in 10 mM PBS (pH 7.4) containing 10% by vol of glycerol, 1 mM reduced glutathione, and 0.2 mM oxidized glutathione and the reaction mixtures were gently stirred at 4 °C for 36 h. The folding of **cK1-1** and **cK1-2** were monitored by LC-MS (Supplementary Figs. 127 and 128). The reaction mixtures were then centrifuged (15,300 × g) at 4 °C during 20 min to eliminate the insoluble aggregates. The mixtures were then transferred in an ultrafiltration column (Vivaspin, 2 mL) with a cut-off of 3000 Da and centrifuged (12,000 × g) at 4 °C. Once concentrated, the folded proteins **cK1-1f** and **cK1-2f** were dialyzed twice (respectively 1 and 18 h) at 4 °C against 10 mM PBS (pH 7.4) containing 10% in vol of glycerol (2 × 0.3 L). Finally, concentrations in **cK1-1f** and **cK1-2f** were determined using BCA assay (Supplementary Table 6). The proteins were analyzed by LC-MS and their disulfide-bond patterns were determined by proteomic analysis using trypsin and endoproteinase N-Asp endopeptidases (Supplementary Methods).

**Biological activity of biotinylated K1 cyclic analogs**. *Alpha screen assays* (Fig. 6d). K1B was produced as described in Simonneau et al.[47]. Recombinant NK1 protein was kindly provided by E. Gherardi (Pavia University, Italy). Competition assays for binding of **K1B**, **cK1-1f**, or **cK1-2f** to recombinant MET-Fc protein (Recombinant human HGFR/c-MET-Fc chimera His-tag protein, carrier free, R&D Systems, 358-MT-100/CF) in competition with increasing concentrations of NK1 protein were performed in 384-well microtiter plates (OptiPlate™-384, PerkinElmer©, CA, USA, 40 μL of final reaction volume). Final concentrations were 20 nM for **K1B**, **cK1-1f**, or **cK1-2f**, 2 nM for MET-Fc, 0–300 nM for NK1, 10 μg mL$^{-1}$ for streptavidin-coated donor beads and protein A-conjugated acceptor beads (AlphaScreen® IgG/protein A detection kit, 6760617C, PerkinElmer). The buffer used for preparing all protein solutions and the bead suspensions was PBS, 5 mM HEPES pH 7.4, 0.1% BSA.

**K1B**, **cK1-1f**, or **cK1-2f** (5 μL, 20 nM) was mixed with a solution of hMET-Fc (5 μL, 2 nM) and with solutions of NK1 (10 μL, 0–300 nM). The mixture was incubated at 23 °C 60 min (final volume 20 μL). Protein A-conjugated acceptor beads (10 μL, 50 μg mL$^{-1}$) were then added to the vials. The plate was incubated at 23 °C for 30 min in a dark box. Finally, streptavidin-coated donor beads (10 μL, 50 μg mL$^{-1}$) were added and the plate was further incubated at 23 °C for 30 min in a dark box. The emitted signal intensity was measured using standard Alpha settings on an EnSpire® Multimode Plate Reader (PerkinElmer). The measurements were in triplicate for each concentration ($n = 3$). The data are expressed as the mean ± standard deviation.

The data were subjected to a nonlinear regression analysis using sigmoidal one-site competition equation with minimal and maximal signals normalized to 0 and 100%, respectively (Sigmaplot software, version 13, Supplementary Fig. 135).

*Western blot analysis of MET activation (Fig. 6e)*. The assay was performed according to Simonneau et al.[47]. HeLa cells were treated for 10 min with 300 pM mature HGF/SF (Recombinant HGF, #PHG0254, Invitrogen), or with 10 nM/100 nM **K1/S**, **cK1-1f/S**, and **cK1-2f/S**, where S stands for streptavidin. Cell lysates were then analyzed by western blot using specific total MET (#37-0100 Invitrogen), total ERK2 (#SC-154 Tebu-bio), phospho-MET (Y1234/1235, clone CD26, #3077 Cell Signaling), phospho-Akt (S473, clone CD9E, #4060 Cell Signaling), phospho-ERK (T202/Y204, clone E10, #9106 Cell Signaling). Cells were collected by scraping and then lysed on ice with a lysis buffer (20 mM HEPES pH 7.4, 142 mM KCl, 5 mM MgCl$_2$, 1 mM EDTA, 5% glycerol, 1% NP40 and 0.1% SDS) supplemented with freshly added protease (1/200 dilution, #P8340, Sigma Aldrich) and phosphatase (1/400 dilution, #P5726, Sigma Aldrich) inhibitors. Lysates were

clarified by centrifugation (20,000 × g, 15 min) and protein concentration was determined (BCA protein assay Kit, Pierce®, Thermo scientific, IL, USA). The same protein amount of cell extracts was separated by NuPAGE gel electrophoresis (4–12% Midi 1.0 mm Bis-Tris precast gels) (Life Technologies) and electrotransferred to polyvinylidene difluoride (PVDF) membranes (Merck Millipore). Membrane was cut between 80 and 110 kDa marker and at 50 kDa to probe simultaneously phospho or total MET, Akt, and ERK. Membranes were probed overnight at 4 °C with primary antibodies diluted to 1/2000 in 5% BSA and 0.1% sodium azide in PBS and followed by incubation with anti-mouse (#115-035-146) or anti-rabbit (#711-035-152) peroxidase-conjugated IgG secondary antibodies (Jackson ImmunoResearch) diluted to 1/30,000 in PBS-casein 0.2%. Protein-antibody complexes were visualized by chemiluminescence with the SuperSignal® West Dura Extended Duration Substrate (Thermo scientific) using a LAS-4000 imaging system (GE HeathCare Life Sciences). The data presented in Fig. 6e are representative of two independent experiments.

*Scattering assays* (Fig. 6f) The assay was performed according to Simonneau et al.[47]. Capan cells were seeded at low density (2000 cells/well on a 12-well plate) to form compact colonies. After treatment, when colony dispersion was observed, the cells were fixed and colored by Hemacolor® stain (Merck, Darmstadt, Germany) according to the manufacturer's instructions. Representative images were captured using a phase contrast microscope with ×40 and ×200 magnification (Nikon Eclipse TS100, Tokyo, Japan). The data presented in Fig. 6f are representative of two independent experiments.

**Reporting summary**. Further information on research design is available in the Nature Research Reporting Summary linked to this article.

## Data availability

The data underlying the findings of this study are available in this article, Supplementary Information, and Source Data files. The source data underlying Figs. 3b, 4b, 6d–f, Supplementary Tables 3–5 and Supplementary Fig. 104 are provided as a Source Data file.

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

## Acknowledgements
We thank ANR for financial support (CyProt, ANR-19CE07-0020).

## Author contributions
V.D. performed the experiments and wrote the manuscript. N.O. prepared the linear K.1. precursor. H.D. performed the proteomic experiments. B.L. and J.V. performed the AlphaScreen® and the cell-based assay. V.A. performed the modelization study and wrote the manuscript. O.M. conceived the study and wrote the manuscript.

## Competing interests
The authors declare no competing interests.
