## [Peer Review File · Nature Communications]

Reviewers' comments:

Reviewer #1 (Remarks to the Author):

The entitled manuscript "A cysteine selenosulfide redox switch for protein chemical synthesis" by Melnyk and coworkers present a nice story for the use of a silent selenoethyl protecting group of N-terminal Cys residue that can be easily released by strong reducing reagent such as TCEP and DTT.

As such the manuscript present a new approach to control the reactivity of cysteine in NCL by introducing N-selenoethyl cysteine (SetCys). The Cys will be free to react only in the presence of TCEP/DTT, MPAA at pH 7.2. The authors found that the selenoethyl arm is removed in water under mild conditions upon the reduction of the selenosulfide bond. The chemistry presented here is quite interesting; yet as this group works only with Cys, because in the case of SetAla, it was orders of magnitude more stable/less reactive. What was missing, in my opinion, is to use this chemistry with Ser, meaning to prepare SetSer analog and test it. Somehow the limitation of this approach make it less interesting, as there are various "protecting groups" of N-terminal Cys residues, although the traceless removal here make it unique.

At first when I saw the chemistry, I thought the N-selenoethyl is used as an auxiliary for NCL reaction, but that would make it much less interesting as it would be used for Cys, the classical residue used for NCL. However, when the N-selenoethyl was shown to be a silent protecting group of the Cys that is removed under very mild conditions, I found it to be interesting, and useful.

Furthermore, the authors presented interesting and convincing kinetic data that support the reaction mechanisms.

However, Figure 4 is confusing. It shows that kinetic product 5 is formed before it is converted into 6 by means of losing the selenoethyl group (Figure 4a), and Figure 4b also show the kinetic data indicating that this step, 5 converted into 6, can take 2 days. Yet, Figure 4c shows that k_5 is very slow, and the peptide 2 is predominantly converted into 3 by losing the selenoethyl arm (this step is r.d.s), and that the conversion of 5 into 6 is not possible under these conditions. The authors should address these concerns, and clarify this important point.

Additionally, I found that HPLC chromatograms that allow us to follow the presented reactions are missing from this paper. Especially important is the HPLC chromatograms for the reactions that shows the intermediates in Figure 2, and 4.

Further, as shown in the proposed mechanism, the reduced species contain free thiol and selenol, more specifically, selenolate, both are capable of reacting with the thioester to provide the ligated product, and in fact, selenolate is a better nucleophile. The author should at least mention this in the text.

The authors should warn the readers that the use of TCEP should always include large excess of ascorbate to prevent any possible deselenization reactions. Did the authors observe this side reaction. If TCEP and DTT are equivalent, perhaps DTT should be superior as it does not cause any side reactions.

I think this work will be more useful and interesting if the authors use the selenoethyl on N-terminal Ser or Thr. The peptide sequence can include an internal Cys. Ser and Thr can promote the proton transfer presented in Figure 3a, and allows the removal of the selenoethyl arm. In this case, the selenoethyl arm can then be called auxiliary allowing NCL at these two residues.

The synthesis of cyclic peptides presented In figure 5 is very nice addition to the paper, and can become useful.

It was not clear to me why the authors did used AcA as protecting group for the synthesis of linear

K1 precursor, why not SetCys?

General things:

- 1) Page 4: ref. 25 is important but irrelevant at its current position.
- 2) Page 5: in the insights into the mechanism of SetCys-mediated ligation, the end of first paragraph(...then slowly disappeared over time in favour..), please add how long it took.
- 3) Page 6: the poor nucleofugality... change nucleophilicity
- 4) Page 8: This result was unexpected because... changing the protein structure will usually have an effect on the activity, and here 10-fold change is not considered massive, in my opinion. I would suggest to re-write this part.
- 5) Please revise the last sentence in manuscript to "more than common additive that are already used in NCL reaction"
- 6) the numbers of the peptides (Figure 2 for example) should be peptide 1 or peptide-1 (numbers not subscript).
- 7) in the SI: as mentioned above the HPLC chromatograms of some of the critical experiments should be included.
- 8) Also the chemical formula of all synthetic small molecules should be included.
- 9) Figure S56 in SI, why the circle points are cut?

Overall it is a nice study, I recommend its acceptance in Nature Communications after major revisions.

Reviewer #2 (Remarks to the Author):

This manuscript from Melnyk and coworkers describes the development of a redox-based selenosulfide protecting group for N-terminal cysteine residues and its application to the synthesis of cyclic peptides. While the SetCys group certainly provides a useful addition to the existing cysteine (and presumably selenocysteine) protecting groups available for use in native chemical ligation, the methodology does not provide any specific benefits over currently available protecting group strategies e.g. thiazolidine. As such, in the opinion of this reviewer the work does not possess the impact or broad applicability that would be necessary for publication in the journal. Below are a number of concerns that would need to be addressed by the authors before publication in another journal:

- 1) The authors do not address the compatibility of the SetCys group with free internal cysteine residues. This is especially important given the reported reduction of the group in the presence of the thiol additive MPAA i.e. would cysteine rich peptides pose a self cleavage risk? This could be tested on a peptide such as a knottin.
- 2) The authors fail to include any crude HPLC chromatograms.ms data of ligations or SetCys deprotections (with the exception of a single trace in figure S66 for the SetCys negative control), giving the reader no indication of the efficiency of either process.
- 3) The characterization of small molecule compounds within the supplementary information file do not meet the standards for publication, with no IR, optical rotation or melting point data currently provided. The authors also do not include the physical state or appearance in any of the synthetic procedures, as is good practice in the area of synthetic organic chemistry.
- 4) The modular strategy described by the authors for the optimization of cyclic peptide topology is of merit, however, the authors fail to describe why the SetCys is essential for such a pursuit; it would seem that many existing methods could be applied to obtain the same scaffolds. The lack of a "selling point" is further compounded by the very small library of cyclic peptides generated in what is purported to be a means of rapidly optimizing linker length i.e. can this method make

these compounds faster, more efficiently etc. While it is not a meaningless pursuit, the authors were not able to recapitulate the native binding affinity with any of the analogues described which also lessened the impact of the overall story.

Reviewer #3 (Remarks to the Author):

Diemer et al. present N-selenoethyl-cysteine (SetCys) as an N-terminal building block for peptide synthesis and as a precursor for cysteine. The selenosulfide version is used as a protected Cys that sheds the selenoethyl group upon reduction at neutral pH in aqueous buffer. The authors exploit these properties for N-terminally protecting peptides during one-pot NCL reactions with several components. In the presence of mildly reducing NCL mediators such as MPAA and another peptide containing a native N-terminal cysteine, no ligation reaction of the SetCys peptide with a peptide thioester is observed. The authors explain this by a kinetic effect that favors the Cys-ligation product over the one with SetCys by 10fold. This kinetic selectivity makes the SetCys residues a remarkable addition to the repertoire of NCL and opens the way for one-pot reactions with finely tuned C-terminal and N-terminal reactivity, e.g. based on pH and addition of reducing agents. The authors provide a detailed kinetic study and sound data on control compounds such as N-mercaptoethyl-cysteine as well as selenoethyl-alanine.

Overall, the authors make a convincing case for the use of SetCys in one-pot, multi-component NCL reactions and demonstrate this for the synthesis of a variety of cyclic peptide for which activity is demonstrated. I believe that the presented data justifies publication in Nature Communications but several points need to be addressed by the authors:

- Provide an explanation why the reactions 1 and 2 in panel 2a proceed as depicted. If the reduced SetCys is formed (postulated for reaction 2) under MPAA conditions at pH 7.2, it must be formed in reaction a as well. Or is TCEP as a Se-scavenger needed?
- Is a 10fold difference in reactivity really sufficient to suppress reaction of the SetCys peptide at similar concentrations?
- Why are no diselenides observed in any of the reactions? Once the reduced SetCys is generated this should happen quickly at concentrations relevant for NCL.
- TCEP often induced deselenization. Has this been observed? This might also be part of the selenoethyl removal reaction as the only observed product TCEP=Se is generated via this pathway as well.
- The authors should early on add more experimental details such as concentrations of peptides and reducing agent.
- What happens with an aryl thioester and a SetCys peptide in the absence of MPAA (or any other thiol) but in the presence of TCEP?
- Why did the authors choose a cyclization reaction where in my opinion a one-pot ligation of three segments would have been more convincing. The cK1 variants are well described and analyzed but only a very selective example. The linear precursor of K1 would have been a nice example.
- Why is ascorbate always present? I understand its use for proving that this is not a radical reaction but why always add it? It often complicates reactions with side reactions. Why was ascorbate absent in controls with N-mercaptoethyl-cysteine?

Reviewer #1 (Remarks to the Author):

The entire manuscript "A cysteine selenosulfide redox switch for protein chemical synthesis" by Melnyk and coworkers present a nice story for the use of a silent selenoethyl protecting group of N-terminal Cys residue that can be easily released by strong reducing reagent such as TCEP and DTT. As such the manuscript present a new approach to control the reactivity of cysteine in NCL by introducing N-selenoethyl cysteine (SetCys). The Cys will be free to react only in the presence of TCEP/DTT, MPAA at pH 7.2. The authors found that the selenoethyl arm is removed in water under mild conditions upon the reduction of the selenosulfide bond. The chemistry presented here is quite interesting; yet as this group works only with Cys, because in the case of SetAla, it was orders of magnitude more stable/less reactive. What was missing, in my opinion, is to use this chemistry with Ser, meaning to prepare SetSer analog and test it. Somehow the limitation of this approach make it less interesting, as there are various "protecting groups" of N-terminal Cys residues, although the traceless removal here make it unique.

At first when I saw the chemistry, I thought the N-selenoethyl is used as an auxiliary for NCL reaction, but that would make it much less interesting as it would be used for Cys, the classical residue used for NCL. However, when the N-selenoethyl was shown to be a silent protecting group of the Cys that is removed under very mild conditions, I found it to be interesting, and useful.

Response:

Thank you very much for highlighting the uniqueness of SetCys and its useful properties.

As discussed in the introduction, the goal of this study was to develop a redox-switch for cysteine, for mimicking what nature does by controlling the reactivity of catalytic cysteines through the establishment of reversible S-S or S-Se bonds.

The concepts put forward in this manuscript are not applicable to amino acids other than Cys for two main reasons.

The first one is due to the mechanism by which the ligation product is formed. Our mechanistic investigations show that the loss of the 2-selenoethyl arm does not occur at the stage of SetCys ligated intermediate **5**. This intermediate must reverse back into a thioester and the starting SetCys peptide, to enable the latter to decompose into Cys peptide which can then ligate irreversibly with the thioester. To further confirm this mechanism, we reacted SetAla peptide with a peptide thioester in the presence of TCEP/MPAA/ascorbate (same conditions as for property **5**, Fig. 2). The reaction yielded at the end the hydrolyzed peptide thioester and Ala peptide with no trace of a ligation product featuring an internal Ala residue.

Even though the method could be applicable to an amino acid other than Cys, we would lose in this case the possibility to block temporarily the 2-selenoethyl arm by an intramolecular Se-S bond as for SetCys, and therefore the redox-switch properties that are sought in this work.

The data for the ligation of SetAla peptide with a model peptide thioester were included in the supporting information (section 6.2.2, new section).

Furthermore, the authors presented interesting and convincing kinetic data that support the reaction mechanisms.

Thank you for the positive comments. The experiment suggested by referee 3 (running the ligation in the absence of ascorbate) further supports the proposed mechanism.

However, Figure 4 is confusing. It shows that kinetic product **5** is formed before it is converted into **6** by means of losing the selenoethyl group (Figure 4a), and Figure 4b also show the kinetic data indicating that this step, **5** converted into **6**, can take 2 days. Yet, Figure 4c shows that k_5 is very slow, and the peptide **2** is predominantly converted into **3** by losing the selenoethyl arm (this step is r.d.s),

and that the conversion of 5 into 6 is not possible under these conditions. The authors should address these concerns, and clarify this important point.

Additionally, I found that HPLC chromatograms that allow us to follow the presented reactions are missing from this paper. Especially important is the HPLC chromatograms for the reactions that shows the intermediates in Figure 2, and 4.

Response:

As drawn Figure 4a was misleading, as it could suggest that 6 is formed directly from 5 as discussed by the referee, which is mechanistically not the case as detailed in Fig. 4b and discussed in the manuscript.

Therefore, Figure 4a was changed to clarify this point and we added also HPLC chromatograms to illustrate the profile of the reaction under kinetic (5 major product) and thermodynamic control (6 major product).

Many HPLCs or LC-MS analysis were included in the Supporting Information file to show the reaction profiles as requested.

Further, as shown in the proposed mechanism, the reduced species contain free thiol and selenol, more specifically, selenolate, both are capable of reacting with the thioester to provide the ligated product, and in fact, selenolate is a better nucleophile. The author should at least mention this in the text.

Response:

We have added this sentence page 5 when we discuss the formation of intermediate 6:

“Of the two nucleophilic sites present in reduced SetCys unit, the selenol moiety is probably the more reactive due to its lower pK_a and higher nucleophilicity”

The authors should warn the readers that the use of TCEP should always include large excess of ascorbate to prevent any possible deselenization reactions. Did the authors observe this side reaction. If TCEP and DTT are equivalent, perhaps DTT should be superior as it does not cause any side reactions.

Response:

We added this sentence when we discuss the mechanism of SetCys to Cys conversion (page 5):

“Omitting ascorbate during the treatment of SetCys peptide 1 by TCEP yields the deselenized peptide Et-CALKEPVHGV-NH₂ as the major product, whose formation competes against the loss of the selenoethyl arm (see Supporting Information).”

The experiment that has been done to document this point can be found in section 4.2 of the Supporting Information (new section).

There is indeed no risk of deselenization with DTT. However, DTT has the disadvantage of providing slower kinetics in comparison to TCEP/ascorbate (see section 4.1 in the Supporting Information). Because ascorbate reduces the deselenization of SetCys to background without side-reactions, we recommend the use of TCEP/ascorbate for the reactions. Note that DTT can lead to side-reactions with peptide thioesters, for example by promoting their conversion to peptide acids.

Nevertheless and to fully respond to the reviewer, we performed the ligation of SetCys peptide 1 with peptide thioester 2 in the presence DTT on a preparative scale. The ligation product was isolated with a yield of 55%, which is slightly less than the yield obtained by using TCEP/ascorbate (64%). This experiment can be found in section 4.7 of the Supporting Information (new section).

I think this work will be more useful and interesting if the authors use the selenoethyl on N-terminal Ser or Thr. The peptide sequence can include an internal Cys. Ser and Thr can promote the proton transfer presented in Figure 3a, and allows the removal of the selenoethyl arm. In this case, the selenoethyl arm can then be called auxiliary allowing NCL at these two residues.

Response:

See our response above to the first comments

The synthesis of cyclic peptides presented in figure 5 is very nice addition to the paper, and can become useful.

Thank you for the positive comments. In addition to this, we have included new results in this figure to show the interest of SetCys chemistry for the synthesis of proteins, in response to referee 3.

It was not clear to me why the authors did use AcA as protecting group for the synthesis of linear K1 precursor, why not SetCys?

Response:

This is because step 3 of the synthetic scheme (section 3.4, Figure S 38 in the supporting information) require TCEP. SetCys would not survive to these conditions.

General things:

- 1) Page 4: ref. 25 is important but irrelevant at its current position.

Response:

We inserted this reference because this review discusses the large difference in reactivity between thiol and selenol compounds, which is what we report also in this manuscript.

We modified the text as follows to be clearer:

“SetCys is a novel illustration of the high difference in reactivity than can exist between thiol and selenol compounds” [ref25]

- 2) Page 5: in the insights into the mechanism of SetCys-mediated ligation, the end of first paragraph(...then slowly disappeared over time in favour...), please add how long it took.

Response:

We replaced “time” by “about two days”

- 3) Page 6: the poor nucleofugality... change nucleophilicity

Response:

We replaced “nucleofugality” by “poor leaving group ability”

- 4) Page 8: This result was unexpected because... changing the protein structure will usually have an effect on the activity, and here 10-fold change is not considered massive, in my opinion. I would suggest to re-write this part.

Response:

We have added a short summary at the end of the paragraph :

“To summarize, cyclization in this case results in a significant loss of affinity, although the backbone cyclization site is opposite to the receptor binding site. However, this loss of affinity does not translate into the cell scattering activity. This work highlights the need for simple synthetic methods toward cyclic proteins to rapidly investigate the interest of backbone cyclization for improving protein properties.”

- 5) Please revised the last sentence in manuscript to "more than common additive that are already used in NCL reaction"

Corrected

- 6) the numbers of the peptides (Figure 2 for example) should be peptide 1 or peptide-1 (numbers not subscript).

Response:

We changed all the figures accordingly, the same for figures of the Supporting Information.

- 7) in the SI: as mentioned above the HPLC chromatograms of some of the critical experiments should be included.

Response:

We included plenty of HPLCs and LC-MS in the Supporting Information.

- 8) Also the chemical formula of all synthetic small molecules should be included.

Response:

All formulas were included initially. This problem might be due to problems with the pdf file.

- 9) Figure S56 in SI, why the circle points are cut?

Response:

This is due to improper pdf conversion.

Overall it is a nice study, I recommend its acceptance in Nature Communications after major revisions.

Thank you very much for the positive comment.

Reviewer #2 (Remarks to the Author):

This manuscript from Melnyk and coworkers describes the development of a redox-based selenosulfide protecting group for N-terminal cysteine residues and its application to the synthesis of cyclic peptides. While the SetCys group certainly provides a useful addition to the existing cysteine (and presumably selenocysteine) protecting groups available for use in native chemical ligation, the methodology does not provide any specific benefits over currently available protecting group strategies e.g. thiazolidine. As such, in the opinion of this reviewer the work does not possess the impact or broad applicability that would be necessary for publication in the journal.

Response:

The design of novel ways of controlling Cys reactivity is a timely challenge as highlighted recently by several works, which include the contributions of Brik's laboratory, see refs 7-10 in the original submission (ref 7: Jbara, M.; Laps, S.; Morgan, M.; Kamnesky, G.; Mann, G.; Wolberger, C.; Brik, A. Palladium prompted on-demand cysteine chemistry for the synthesis of challenging and uniquely modified proteins. *Nat. Commun.* 2018, 9 (1), 3154).

These works are primarily conducted to overcome the known limitations of existing protecting groups for Cys, including thiazolidine which is not devoid of drawbacks. We discussed these points very recently in a review (Agouridas, V. *Chem. Rev.* 2019).

The chemical behavior of SetCys is more subtle than a classical protected Cys residue.

SetCys reacts under weak reducing conditions (Fig. 2a), which is not really what one expects from a protected Cys residue according to the classical definition.

When a Cys peptide and a SetCys peptide are present simultaneously in a solution and react with a peptide thioester, the Cys peptide reacts faster but SetCys reacts too, reversibly. Thus, even if the SetCys ligation product is formed, it goes back into reactants. The thioester ends up being completely captured by the Cys peptide because this process acts as a sink by being irreversible.

To clarify these aspects in response to comments from referee 3, we modelled the competitive ligation process. The results of this study can be found in the annex placed at the end of this response letter. The model suggests that both ligation products are formed at the beginning of the reaction albeit in favor of the ligation product from the Cys peptide, but the product from SetCys peptide declines over time.

SetCys is a redox-switch for cysteine that enables a unique chemistry and requires nothing more than common and harmless additives that are already used in NCL reactions.

Below are a number of concerns that would need to be addressed by the authors before publication in another journal:

- 1) The authors do not address the compatibility of the SetCys group with free internal cysteine residues. This is especially important given the reported reduction of the group in the presence of the thiol additive MPAA i.e. would cysteine rich peptides pose a self cleavage risk? This could be tested on a peptide such as a knottin.

Response:

We thank the referee for this interesting suggestion. We successfully synthesized a Cys-rich peptide, conotoxin OIVA, by using a peptide segment featuring an N-terminal SetCys residue with a Cys residue adjacent to it.

To mention this, the following sentence was included in the main manuscript:

« We also verified that internal Cys residues are unable to activate SetCys residue, which is therefore useful for the production of Cys-rich peptides (see Supporting Information).”

The data now constitute section 4.4.3 (new section) in the Supporting Information.

- 2) The authors fail to include any crude HPLC chromatograms.ms data of ligations or SetCys deprotections (with the exception of a single trace in figure S66 for the SetCys negative control), giving the reader no indication of the efficiency of either process.

3)

Response:

We have included plenty of HPLCs and LC-MS in the supporting Information, and also in Fig. 4.

- 4) The characterization of small molecule compounds within the supplementary information file do not meet the standards for publication, with no IR, optical rotation or melting point data currently provided. The authors also do not include the physical state or appearance in any of the synthetic procedures, as is good practice in the area of synthetic organic chemistry.

5)

Response:

IR and optical rotation are now provided in the Supporting Information. Products are not crystalline.

4) The modular strategy described by the authors for the optimization of cyclic peptide topology is of merit, however, the authors fail to describe why the SetCys is essential for such a pursuit; it would seem that many existing methods could be applied to obtain the same scaffolds. The lack of a “selling point” is further compounded by the very small library of cyclic peptides generated in what is purported to be a means of rapidly optimizing linker length i.e. can this method make these

compounds faster, more efficiently etc. While it is not a meaningless pursuit, the authors were not able to recapitulate the native binding affinity with any of the analogues described which also lessened the impact of the overall story.

Response:

K1 domain is essentially made of loops as shown in Fig. 6a. What comes immediately to mind when looking at our data is that a conformational change might occur within the K1 domain upon binding to the receptor, which could be disfavored in the cyclic analogs.

Up to now, all attempts for obtaining high resolution X-ray crystallographic data for K1-MET complex failed. We are collaborating on this topic with the team of Pr Ermanno Gherardi (Univ Pavia, Italy), the co-discoverer of HGF/SF. Thus, at present we cannot compare the conformation of K1 alone and in complex with MET and thus we cannot predict the impact of cyclization. In this context, any information that provide new clues on structure-activity relationships for K1 HGF/SF domain is welcome.

Therefore, we think that our observations are interesting i) for the community working on the structural biology HGF/SF, ii) for the researchers working on protein cyclization, that must be aware that cyclization is not always beneficial as one would expect.

The difficulty in predicting the outcome of protein cyclization reinforces the need for simple methods enabling the access to cyclic proteins. SetCys is clearly a unique tool for that, and the one-pot cyclization method described in Figure 5 has no equivalent in the literature. Making more K1 cyclic analogs, especially after considering the above mentioned arguments, will not change the conclusion that SetCys chemistry is conceptually novel and extremely useful. We believe that using SetCys for preparing Cys-rich peptides as suggested by this referee, or linear proteins as suggested by referee 3, is more appropriate for effectively complete and enlarge the scope of the work.

Reviewer #3 (Remarks to the Author):

Diemer et al. present N-selenoethyl-cysteine (SetCys) as an N-terminal building block for peptide synthesis and as a precursor for cysteine. The selenosulfide version is used as a protected Cys that sheds the selenoethyl group upon reduction at neutral pH in aqueous buffer. The authors exploit these properties for N-terminally protecting peptides during one-pot NCL reactions with several components. In the presence of mildly reducing NCL mediators such as MPAA and another peptide containing a native N-terminal cysteine, no ligation reaction of the SetCys peptide with a peptide thioester is observed. The authors explain this by a kinetic effect that favors the Cys-ligation product over the one with SetCys by 10fold. This kinetic selectivity makes the SetCys residues a remarkable addition to the repertoire of NCL and opens the way for one-pot reactions with finely tuned C-terminal and N-terminal reactivity, e.g. based on pH and addition of reducing agents. The authors provide a

detailed kinetic study and sound data on control compounds such as N-mercaptoethyl-cysteine as well as selenoethyl-alanine.

Overall, the authors make a convincing case for the use of SetCys in one-pot, multi-component NCL reactions and demonstrate this for the synthesis of a variety of cyclic peptide for which activity is demonstrated. I believe that the presented data justifies publication in Nature Communications but several points need to be addressed by the authors:

- Provide an explanation why the reactions 1 and 2 in panel 2a proceed as depicted. If the reduced SetCys is formed (postulated for reaction 2) under MPAA conditions at pH 7.2, it must be formed in reaction a as well.

Response:

The difficulty in detecting the reduced SetCys peptide in reaction 1 or 2 (Figure 2A) comes probably from the low concentrations for this species in solution and also its ease of oxidation into SetCys during workup.

We have removed the formula from Fig 2a because the formation of the

SetCys amide product might also involve intermediate as well.

- Is a 10fold difference in reactivity really sufficient to suppress reaction of the SetCys peptide at similar concentrations?

Response:

The question is very interesting because discussing this point requires to go deeply into the intimate steps involved in SetCys chemistry.

The assertion that a classical NCL proceeds at a ~10-fold higher apparent rate than a SetCys ligation in weak reducing conditions is perfectly valid. However, the direct comparison between apparent kinetic rates is in fact misleading. A classical NCL is under the control of thiol-thioester exchanges. In contrast, the rate of a SetCys ligation under weak reducing conditions is limited by the reduction of the selenosulfide bond. In the latter case, this can be seen through the early accumulation of the aryl thioester in the reaction mixture. In a competitive experiment, the transient aryl thioester formed would thus be immediately consumed by the Cys peptide, while the SetCys peptide would barely be reduced.

In the annex placed at the end of this response letter, we have modelled the competitive experiment using the kinetic data we have in hand. We are aware that the proposed model does not account for all the chemical processes occurring in the reaction. In particular, working under low reducing conditions implies necessarily that part of the thiol and selenol moieties are engaged in mixed dichalcogenides. Determination of the rate constants for these species is extremely difficult given their low concentration in solution and their ease of oxidation during workup. Nevertheless, we consider that the information provided by the model is qualitatively correct in that the origin of the high selectivity stems from the reversibility of SetCys amide product formation.

Because of the limitations discussed above regarding the proposed model for the competitive ligation experiment, the description of the model and its discussion was not inserted in the Supporting Information. However, we think that this discussion is informative and we will be happy to add it to the Supporting Information if this referee considers that it is worth to do so.

- Why are no diselenides observed in any of the reactions? Once the reduced SetCys is generated this should happen quickly at concentrations relevant for NCL.

Response:

Diselenides are reduced by TCEP (or DTT) under strong reducing conditions.

The reducing potential of selenosulfides is pretty close to those of diselenides. But 7-membered ring dichalcogenides are much more resistant to reductants than acyclic dichalcogenides. Thus, under

weak reducing conditions, oxidation of reduced SetCys, i.e. , likely favors the formation of the SetCys cyclic selenosulfide rather than acyclic diselenides.

- TCEP often induced deselenization. Has this been observed? This might also be part of the selenoethyl removal reaction as the only observed product TCEP=Se is generated via this pathway as well.

Response:

To document this point, we treated SetCys peptide with TCEP in the absence of ascorbate. The results of this experiment now constitutes section 4.2 of the Supporting Information.

TCEP-induced deselenization occurs massively if ascorbate is omitted from the reaction mixture. The deselenization process yields an *N*-ethyl cysteine peptide as by-product, whose formation competes with the formation of the Cys peptide.

Omitting ascorbate from the reaction mixture is detrimental to the SetCys to Cys conversion. This experiment further supports the ionic mechanism we propose for the SetCys to Cys conversion.

- The authors should early on add more experimental details such as concentrations of peptides and reducing agent.

Corrected

- What happens with an aryl thioester and a SetCys peptide in the absence of MPAA (or any other thiol) but in the presence of TCEP?

Response:

We thank the referee for this interesting suggestion.

We performed the reaction of an aryl thioester with a SetCys in the presence of TCEP/ascorbate but in the absence of MPAA as proposed.

When SetCys peptide 1 and peptidyl MPAA-thioester ILKEPWHGA-MPAA were reacted in the presence of the TCEP, native peptide ILKEPWHGA-CALKEPVHGV-NH₂ was recovered as the only ligation product with 40% yield.

The results of this study were inserted in the Supporting Information and constitute section 4.6 (novel section).

-Why did the authors choose a cyclization reaction where in my opinion a one-pot ligation of three segments would have been more convincing. The cK1 variants are well described and analyzed but only a very selective example. The linear precursor of K1 would have been a nice example.

Response:

The powerfulness of SetCys chemistry is exploited at best in the production of cyclic proteins from large linear precursors, explaining why we chose this type of application.

We agree with the referee that the use of SetCys for the production of linear proteins such as K1 would significantly enlarge the scope of the study.

In fact, we performed the synthesis of K1 linear polypeptide by two different methods (Figure 5c in the main manuscript). In the first method (method A), the peptide segments were added sequentially. In the second method (method B), all three peptide segments were mixed at the beginning of the assembly process.

We inserted the following sentences in the main manuscript to describe the work:

“SetCys chemistry proved equally useful for the C-to-N one-pot assembly of linear polypeptides from three peptide segments (Figure 5a,c). The peptide segments could be added sequentially (Figure 5c, Method A) or mixed altogether from the beginning of the assembly process (Figure 5c, Method B) with equal selectivity and efficiency.”

The experiments that correspond to these syntheses can be found in section 9 in the Supporting Information (new section).

- Why is ascorbate always present? I understand its use for proving that this is not a radical reaction but why always add it? It often complicates reactions with side reactions.

Response:

To document this point, we treated SetCys peptide with TCEP in the absence of ascorbate. The results of this experiment constitutes now section 4.2 of the Supporting Information.

TCEP-induced deselenization occurs massively if ascorbate is omitted from the reaction mixture. The deselenization process yields an *N*-ethyl cysteine peptide as by-product, whose formation competes with the formation of the Cys peptide.

Omitting ascorbate from the reaction mixture is detrimental to the SetCys to Cys conversion. This experiment further supports the ionic mechanism we propose for the SetCys to Cys conversion.

Why was ascorbate absent in controls with N-mercaptoethyl-cysteine?

Response:

MPAA is an excellent quencher of thiyl radicals, and thus of the TCEP-induced free-radical based desulfurization reaction. Therefore, ascorbate is not needed in this case.

In contrast, selenyl radicals form so easily that MPAA and ascorbate are needed to avoid any free radical-based deselenization process.

Annex

Discussion of the origin of the observed selectivity

Our experimental results show that an NCL reaction conducted under weak reducing conditions (200 mM MPAA) operates ~10 times faster than a SetCys ligation at similar peptide concentration. How such a difference in apparent rates could explain the high selectivities observed in a competitive ligation experiment? At first glance, with such apparent rates one would expect roughly a 90/10 distribution of products. Experimentally, this is not the case as the ligation product between the peptide thioester and the SetCys peptide was not detected in our experiments.

In order to gain insight into the origin of the selectivity in the competitive ligation process, we modelled the competitive ligation in an attempt to estimate the distribution of the products using the kinetic data in our hand (Figure S 1). We are aware that the model does not account for all the chemical processes occurring in the reaction. Working under low reducing conditions implies necessarily that part of the thiol and selenol moieties are engaged in mixed

dichalcogenides. This is especially true for SetCys peptide 2 and SetCys ligation product 5 in the model depicted in Figure S 1. Nevertheless, we consider that the information provided by the model is qualitatively correct in that the origin of the high selectivity stems from the reversibility of SetCys amide product 5 formation.

Figure S 1. Mechanistic model for the competitive NCL of SetCys and Cys peptides in the presence of 200 mM MPAA.

According to our model, peptide alkyl thioester 4 undergoes thiol-thioester exchanges to form a transient aryl thioester 7. Once formed, the latter is trapped by either Cys peptide 3 or reduced SetCys peptide 2. The cleavage of the selenosulfide bond of SetCys peptide 1 by MPAA to form reduced SetCys peptide 2 is associated to an apparent rate constant k_{redapp} . While ligation of Cys peptide 3 is irreversible, ligation of reduced SetCys peptide 2 leading to the formation of SetCys amide 5 is reversible. Of note is that reversible oxidation of peptide 5 by formation of an intramolecular selenosulfide bond was not considered in this model.

In Figure S 1, the k_{+2} , k_{-2} , k_{+3} , k_{-3} rate constants refer to the previously determined rate constants in the mechanistic model of the SetCys ligation (see Figure 4 in the main manuscript), which were reused for modelling the SetCys ligation process under weak reducing conditions.

To determine k_{redapp} , we modelled the ligation reaction of SetCys peptide 1 with thioester 4 in the presence of 200 mM MPAA (Figure S 2, see also section 3.2.1 Native Chemical Ligation performed in low reducing conditions (absence of TCEP)). Fitting of the reaction enabled us to determine a value for k_{redapp} ($0.0030 \text{ M}^{-1} \cdot \text{min}^{-1}$). This value implies that the reduction of

the selenosulfide bond is the rate determining step of the process. This hypothesis is supported experimentally by the fact that under low reducing conditions peptide aryl thioester **7** accumulates to a significant extent in the first minutes of reaction, whereas it is readily consumed under strong reducing conditions in the same peptide concentration range.

B)

Figure S 2. A) Mechanistic model for the NCL of SetCys peptides in the presence of 200 mM MPAA. B) Ligation reaction between RLKEPVHGA-MPA (8 mM) and SetCys-ALKEPVHGV-NH₂ (9.2 mM) at pH 7.2 in the presence of 200 mM MPAA used as a reducing agent. The red dots correspond to the conversion of starting peptides to ligated product RLKEPVHGA-SetCys-ALKEPVHGV-NH₂. The continuous red curve corresponds

to the fitting of the reaction using Kintek Explorer Software™ (Version 7.2.180216., Kintek Corporation - <https://kintekcorp.com/software/>) to the model described in A).

Having determined a value for $k_{\text{red,app}}$, numerical integration of rate equations allowed us to determine a theoretical conversion profile for ligated peptides 5 and 6 (Figure S 3). Noteworthy, the model predicted the almost exclusive formation of peptide 6 for reaction times $> \sim 700$ min, which supports the experimental observations. This modelization clearly shows that although SetCys amide 5 might form at the beginning of the reaction, its proportion in the mixture declines over times because its formation is reversible.

Figure S 3. Simulated conversion profile of peptides 1 and 3 into peptides 5 and 6 respectively in a competitive native chemical ligation reaction (200 mM MPAA, pH 7.2).

REVIEWERS' COMMENTS:

Reviewer #1 (Remarks to the Author):

the authors have addressed all the concerns and revised the manuscript according to the referees reports. I suggest its acceptance as is.

Reviewer #2 (Remarks to the Author):

In their revision the authors have highlighted the nuanced behavior of the SetCys system but have failed to respond to the key concern of how it provides any practical benefit to the community, especially in direct comparison to existing Cys protecting groups; the authors suggest that there are problems with Thz yet this is frequently and successfully used for protein synthesis. There are a raft of other Cys protecting groups that can and have been used to facilitate ligation chemistry (as well as other chemistry). I note that the authors do not mention any previously reported Cys protecting groups in the introduction of the manuscript or highlight any additional or complementary functionality of the SetCys system in comparison to established methods. Simply reporting a new protecting group which demonstrates esoteric behavior but no functional advantage over existing methods does not have the impact for publication in Nature Communications (in this reviewer's opinion).

The authors cite the work of Jbara et al. as an example of a comparable study published in Nature Communications, however, the Pd-mediated deprotection reported in this work showcases the tunable reactivity across three protecting groups, together with the assembly of large proteins from multiple fragments in a controlled manner. Neither the chemoselectivity nor synthetic utility demonstrated by Jbara et al. is present in this work.

The authors chose not to address the concerns over the small library of cyclic peptides accessed. To clarify this was not a concern over the number of analogues made for a specific target, but rather the claim that this represents a generic, high throughput platform for the optimization of linker length. By focusing on a single system, the authors do not test the scope of the SetCys system which in the opinion of this reviewer would be more impactful.

Reviewer #3 (Remarks to the Author):

Diemer et al. have satisfactorily addressed all critical points raised by three reviewers in the first round of reviewing.

I have only a minor point to be addressed regarding the newly added data before publication of this manuscript in Nature Communications:

The new data in Figure 5c on the synthesis of linear K1 should be slightly changed as the peptide sequences (12-14) are missing the cysteine (or Set) residues required for ligation reactions.

Dear reviewers,

We would like to thank you again for your comments that helped us further challenge the SetCys approach, enrich its scope and improve the manuscript.

Best Regards

Oleg Melnyk and Vangelis Agouridas

REVIEWERS' COMMENTS:

Reviewer #1 (Remarks to the Author):

the authors have addressed all the concerns and revised the manuscript according to the referees reports. I suggest its acceptance as is.

Thank you.

Reviewer #2 (Remarks to the Author):

In their revision the authors have highlighted the nuanced behavior of the SetCys system but have failed to respond to the key concern of how it provides any practical benefit to the community, especially in direct comparison to existing Cys protecting groups; the authors suggest that there are problems with Thz yet this is frequently and successfully used for protein synthesis. There are a raft of other Cys protecting groups that can and have been used to facilitate ligation chemistry (as well as other chemistry). I note that the authors do not mention any previously reported Cys protecting groups in the introduction of the manuscript or highlight any additional or complementary functionality of the SetCys system in comparison to established methods. Simply reporting a new protecting group which demonstrates esoteric behavior but no functional advantage over existing methods does not have the impact for publication in Nature Communications (in this reviewer's opinion).

The following sentence was in the introduction with the aim to direct the reader to an exhaustive review we published last year in Chemical Reviews and which describes all the protecting groups used for protein synthesis:

“In practice, Cys reactivity is instead masked during protein assembly by introducing classical alkyl- or acyl-based protecting groups on the α -amino group, on the side-chain thiol or both (for a recent review see reference ²).”

This sentence was modified to mention the thiazolidine protection for cysteine, which is certainly the most popular in the field:

In practice, Cys reactivity is instead masked during protein assembly by introducing classical alkyl- or acyl-based protecting groups on the α -amino group, on the side-chain thiol or both. Among the various cysteine protection strategies described so far (for a recent review see reference ²), thiazolidine protecting group is certainly the most popular for protein chemical synthesis.¹⁵

The authors cite the work of Jbara et al. as an example of a comparable study published in Nature Communications, however, the Pd-mediated deprotection reported in this work showcases the tunable reactivity across three protecting groups, together with the assembly of large proteins from multiple fragments in a controlled manner. Neither the chemoselectivity nor synthetic utility demonstrated by Jbara et al. is present in this work.

The authors chose not to address the concerns over the small library of cyclic peptides accessed. To clarify this was not a concern over the number of analogues made for a specific target, but rather the claim that this represents a generic, high throughput platform for the optimization of linker length. By focusing on a single system, the authors do not test the scope of the SetCys system which in the opinion of this reviewer would be more impactful.

Reviewer #3 (Remarks to the Author):

Diemer et al. have satisfactorily addressed all critical points raised by three reviewers in the first round of reviewing.

Thank you.

I have only a minor point to be addressed regarding the newly added data before publication of this manuscript in Nature Communications:

The new data in Figure 5c on the synthesis of linear K1 should be slightly changed as the peptide sequences (12-14) are missing the cysteine (or Set) residues required for ligation reactions.

These sequences are deliberately missing the Cys or SetCys residues as dedicated symbols added at the N-terminus of the thick lines are used for their representation (black, red, green dots or black circles, see figure 1a).

However, the figure has been modified following your suggestions and now includes Cys or SetCys residues in the sequence of the peptidic segments, as well as the C-terminal carboxy derivative (thioester or SEA-amide).